# Plant-Derived Monoterpene Therapies in Parkinson’s Disease Models: Systematic Review and Meta-Analysis

**DOI:** 10.3390/plants14070999

**Published:** 2025-03-22

**Authors:** Matías Jávega-Cometto, Aracely J. Naranjo-Viteri, Leandro G. Champarini, Claudia B. Hereñú, Rosana Crespo

**Affiliations:** Instituto de Farmacología Experimental de Córdoba (IFEC-CONICET), Departamento de Farmacología Otto Orsingher, Facultad de Ciencias Químicas, Universidad Nacional de Córdoba, Córdoba X5000HUA, Argentina; matias.javega@unc.edu.ar (M.J.-C.); aracely.naranjo.viteri@mi.unc.edu.ar (A.J.N.-V.); leandro.champarini@unc.edu.ar (L.G.C.); claudia.herenu@unc.edu.ar (C.B.H.)

**Keywords:** monoterpenes, Parkinson’s animal model, behavioral tests, oxidative stress, neuroinflammation

## Abstract

Monoterpenes (MTs) are plants’ secondary metabolites and major components of essential oils (EOs), widely used in the pharmaceutical industry. However, its neuroprotective effects, particularly in Parkinson’s disease (PD) have not been fully demonstrated. PD is a progressive neurological disorder marked by dopaminergic neuron loss in the substantia nigra, motor symptoms being the most reported ones. This review evaluates the evidence supporting the use of MTs as potential neuroprotective agents. PubMed, SCOPUS, Google Scholar, and ScienceDirect databases were searched for articles on MTs in murine models with any type of administration. The PRISMA guidelines were followed. After screening 405 records, 32 were included in the systematic review and 30 were included in the meta-analysis. Fifteen MTs, commonly found in EOs, were identified as potential therapeutic agents for PD. The meta-analysis revealed that MTs administration improved motor performance, increased tyrosine hydroxylase levels, reduced oxidative stress markers (malondialdehyde) and proinflammatory cytokines (IL-6, IL-1, TNF-α), and enhanced antioxidant enzymes (catalase, superoxide dismutase) in parkinsonian animals. The antioxidant and anti-inflammatory properties of MTs appear to be key mechanisms in mitigating dopaminergic neurodegeneration. However, further clinical research is essential to translate these findings into practical applications.

## 1. Introduction

Monoterpenes (MTs) are the main natural volatile organic components of plant essential oils (EOs). They are responsible for the characteristic aroma of many plants and have a basic structure that consists of two isoprene units, having in total ten carbon atoms. Modified MTs, such as those with oxygen-containing functional groups or missing a methyl group, are also called monoterpenoids. MTs have a great variety of acyclic or cyclic structures, resulting from changes within the isoprenoid chain through chemical reactions, including reductions, oxidations, cyclizations, ring cleavages, or rearrangements [1,2]. Although they are considered secondary metabolites because they are not essential for the plant’s viability, they fulfill an important role in the plant defense system against biotic and/or abiotic stress [3,4,5].

MTs are important bioactive compounds [6], with a wide range of biological activities, including antifungal, antibacterial, antiviral, antioxidant, anticancer, antidiabetic, antiarrhythmic, local anesthetic, antinociceptive, anti-inflammatory, antihistaminic, and antispasmodic activities, among others [7,8]. In vitro and in vivo studies have shown that their underlying mechanisms of action primarily involve the reduction of oxidative stress through their free radical scavenging ability. Additionally, MTs enhance endogenous defense mechanisms by modulating the activity of antioxidant enzymes such as superoxide dismutase (SOD), glutathione-related enzymes, and catalase (CAT), which neutralize pro-oxidant molecules [9]. The modulation of various protein factors involved in cellular signaling pathways—such as those regulating apoptosis, cell division, autophagy, and inflammation—has been reported, highlighting their pleiotropic effects. Among these, their neuroprotective properties have been evaluated, particularly in mitigating oxidative damage and microglia-mediated inflammation [10,11], both commonly implicated in neurodegenerative processes [12,13].

Parkinson’s disease (PD) is the second most common neurodegenerative disorder, characterized by the progressive degeneration of dopaminergic neurons in the substantia nigra (SN), a brain region critical for motor control, due to intracellular α-synuclein inclusions. Within dopaminergic neurons, dopamine (DA) synthesis is catalyzed by tyrosine hydroxylase (TH). TH is released into the synaptic cleft where DA spreads by diffusion. DA interacts with specific receptors and is eliminated through reuptake into presynaptic terminals by DA transporter (DAT). Then, DA is storaged in synaptic vesicles through the vesicular monoamine transporter 2 (VMAT2) [14,15]. These components of the dopaminergic neuron and synapse (dopamine-related parameters) are widely measured in SN and striatum (CPU) to determine the integrity of DA neurons and the severity of the neurodegeneration. Because the axons of SN neurons arborize profusely within CPU [16], TH and DAT are reliable markers of terminal dopamine innervation in the brain structure [17]. The dopaminergic neurons’ degeneration leads to a reduction in DA levels in the brain and is closely associated with neuroinflammation. Neuroinflammation involves the activation of immune cells which release pro-inflammatory cytokines (IL-1, IL-6, TNF-α) and reactive oxygen species (ROS), which accelerate nerve cell degeneration [18,19]. Moreover, the impaired mitochondrial function also leads to increased ROS production, activation of death signaling pathways, and deficits in neuronal homeostasis [20,21,22].

Since current therapies for PD have limitations in terms of their long-term efficacy and side effects, the search for plant-derived natural compounds as potential therapeutic agents has significantly increased in recent years. EOs have been used in traditional medicine in very ancient cultures. MTs stand out as the main bioactive components in EOs, being present in countless combinations whose interactions (synergistic, additive, and/or inhibitory) can produce variable effects according to the Eos’ composition [23,24]. For this reason, studying the effects and mechanisms of action of MTs provides valuable insights into their potential therapeutic use. In order to make advances in the knowledge of the application of MTs in PD, the meta-analysis of existing data shall prove to be a useful approach to integrate findings and construct a more comprehensive understanding of the matter. Furthermore, the analysis of MTs’ administration in preclinical animal models provides information that enhances their translation to clinical studies. Murine models of PD are widely used to study the neurodegenerative nigrostriatal process [25], as they exhibit behavioral, biochemical, and molecular neuroinflammatory alterations like those observed in humans. These models also demonstrate dopaminergic neuron loss driven by the formation of various oxidants and free radicals, neuroinflammation, lipid peroxidation, and the depletion of reduced glutathione. Despite its limitations in brain structures, murine models remain a valuable tool for investigating the pathogenesis and progression of PD [26].

The objective of this study was to compile and conduct a comparative statistical analysis of published results on the effects of MTs found in EOs on parkinsonian animals (PDAs) and, in turn, to evaluate their potential as therapeutic agents to inhibit, reduce, or delay neuronal and symptomatic deterioration in PD patients. This research aimed to present current information on the effects and possible mechanisms of action of MTs in preclinical PD models, contributing to the development of new phytotherapies.

## 2. Results

### 2.1. Search Results

Figure 1 shows the results of the literary search. After screening 405 records for their titles and abstracts, and 130 for their full text, 32 were included in the systematic review. After extracting the data from the publications, 30 studies were included in the meta-analysis since two of them did not report results that could be used for the analysis per the established criteria.

### 2.2. Characteristics of Publications

Table 1 summarizes the extracted data from the publications included in this systematic review. The 32 studies encompassed 15 MTs: Borneol, Carvacrol, Citronellol, Geraniol, Limonene, Linalool, Menthol, Myrcene, Myrtenal, Myrtenol, Perillyl Alcohol, α-Pinene, Safranal, Thymol, and Thymoquinone. The majority of studies (22) tested a cyclic monoterpenoid, 8 worked with an acyclic monoterpenoid, and only 2 studies tested an aliphatic monoterpene (one cyclic and one acyclic). The most utilized routes of administration were oral (56%) and intraperitoneal (41%), with only one publication reporting intragastric administration. Because of the hydrophobic nature of most of the compounds, the inclusion of the vehicle solution or emulsification strategy was also reported. The most commonly used vehicles included Tween 80 and various oils to emulsify the MTs solution and enhance its bioavailability. Regarding the timing of administration, 46% of the protocols administered the MTs concurrently with the neurotoxic compounds, 31% administered them beforehand to assess preventive effects, and 23% administered them afterward.

Out of the 32 articles selected, 65% correspond to studies conducted in rats, primarily Wistar (56%) and Sprague–Dawley (6%). The remaining 35% of the selected studies were conducted in C57BL/6 mice (29%) and Swiss mice (6%). All experiments were carried out on adult male animals. All parkinsonism models in this study were chemically induced. Common inducers of PD used in these publications included 6-hydroxydopamine (6-OHDA) (35% of the protocols), 1-methyl-4-phenyl-1,2,3,6-tetrahydropyridine (MPTP) (24%), rotenone (21%), and reserpine (9%). Other non-common neurotoxic compounds used included lipopolysaccharides (LPS), haloperidol, trichloroethylene, and Mn^2+^. All these publications reported motor deficits and/or dopaminergic depletion in response to the administration of the neurotoxins.

Although we focused on the in vivo experiments of the analyzed publications, many of them also include in vitro experiments that complement their findings in the animal models.

Table 1 Summary of data extracted from publications. ↑ (statistically significant increase); ↓ (statistically significant decrease); ≈ (non-significant statistical difference); p.o. (oral administration); i.p. (intraperitoneal administration); s.c. (subcutaneous administration); MPTP (1-methyl-4-phenyl-1,2,3,6-tetrahydropyridine); 6-OHDA (6-hydroxydopamine); LPS (Lipopolysaccharides); MFB (medial forebrain bundle); SN (substantia nigra pars compacta); CPU (striatum); VTA (ventral tegmental area); Hipp (Hippocampus); PFC (Prefrontal Cortex); CSF (Cerebrospinal fluid); TH (tyrosine hydroxylase); DAT (dopamine active transporter); VMAT2 (vesicular monoamine transporter-2); DA (dopamine); DOPAC (3,4-Dihydroxyphenylacetic acid); HVA (Homovanillic acid); MAO-B (monoamine oxidase-B); ROS (Reactive Oxygen Species); MDA (malondialdehyde); SOD (superoxide dismutase); GSH (glutathione); GSH-px (glutathione peroxidase); NRF2 (Nuclear factor erythroid 2-related factor 2); NQO-1 (NAD(P)H dehydrogenase (quinone) 1); OH-1 (Heme oxygenase 1); 8OH-dG (8-hydroxydeoxyguanosine); GST (Glutathione S-transferase); IL-6 (Interleukin 6); IL-1β (Interleukin 1 beta); TNF-α (Tumor Necrosis Factor); IL-4 (Interleukin 4); COX-2 (Cyclooxygenase-2); iNOS (inducible Nitric Oxide Synthase); IBA1/Aif (Ionized Calcium-binding Adapter molecule 1/Allograft Inflammatory Factor 1); GFAP (Glial Fibrillary Acidic Protein); NLRP3 (NLR family Pyrin domain containing 3); ASC (Apoptosis associated Speck-like protein); MMP-9 (Matrix Metalloproteinase-9); (p-)NFκB ((phosphorylated)-Nuclear Factor Kappa B); p-IκB ((phosphorylated)-Inhibitor of nuclear factor Kappa B); TLR4 (Toll-Like Receptor 4); TRPC (Transient Receptor Potential Cation channel); TRPA1 (Transient Receptor Potential Ankyrin 1); mTOR (mammalian Target Of Rapamycin); LC-3 (Microtubule-associated protein 1A/1B-light chain 3); BDNF (Brain-Derived Neurotrophic Factor); GDNF (Glial cell line-Derived Neurotrophic Factor); Cyc-1 (Cytochrome c1); JNK (c-Jun N-terminal Kinases); CHOP (C/EBP Homologous Protein); LC3B (Microtubule-associated proteins 1A/1B light chain 3B); PGC-1α (Peroxisome proliferator-activated receptor Gamma Coactivator 1-alpha); DRP1 (Dynamin-Related Protein 1); NE (Norepinephrine); 5-HT (5-Hydroxytryptamine); GABA (Gamma-aminobutyric Acid); ACHE (Acetylcholinesterase).

### 2.3. Risk of Bias

Figure 2 shows a graph of the overall risk of bias of each domain. We represent the percentage of publications included in the meta-analysis whose risk of bias was low (green), moderate (yellow), or critical (red). According to our analysis, 50% of publications included in the meta-analysis showed low risk of selection bias, while the other 50% showed moderate risk. All publications presented moderate risk of performance bias and detection bias. A high risk of attrition bias was predominant (63%), while 34% had moderate risk and only 3% of publications had low risk of bias. On the other hand, 93% of publications had a low risk of reporting bias, while 7% had moderate risk. Other risks of bias were considered moderate in 97% of publications and high in 3%. Overall, 14% of publications had a low risk of bias while the other 86% had a moderate risk.

**Table 1 plants-14-00999-t001:** Description of the main characteristics of preclinical studies (murine in vivo studies) using monoterpenes for Parkinson’s disease.

Monoterpene Therapy	Animal Model Characteristics	Motor Behavior	Dopamine- Related Parameters	Antioxidant Properties	Inflammatory Parameters	Other Findings	Reference
Borneol (i.g. for 30 days before lesion. 7.5 mg/kg; 15 mg/kg; 20 mg/kg; 30 mg/kg)	Male C57BL/6 Mouse; 7 weeks old. MPTP i.p. 20 mg/kg for 30 days	↑ Distance traveled in Open Field ↑ Performance in Pole Climbing ↑ Time in Suspension Test	SN: ↑ TH positive cells, ↑ DAT positive cells. STR: ↑ TH positive fibers, ↑ DAT positive fibers, ↑ DAT protein and mRNA content, ↑ VMAT2 protein and mRNA content, ↑ DA, ↑ DOPAC, ↑ HVA	SN: ↓ MDA, ↑ SOD, ↑ GSH-px	SN: ↓ IL-6, ↓ IL-1β, ↓ TNF-α	↓ Freezing in Open Field	Ding et al., 2024 [27]
Carvacrol (i.p. for 14 days after lesion. 10 mg/kg)	Male Sprague-Dawley Rat; 200–250 g. 6-OHDA (12 μg, intrastriatal, unilateral)		SN: ↑ TH positive cells		SN: ↓ GFAP positive cells	SN: ↑ TRPC positive DA neurons. ↓ TRPC positive astrocytes. ↑ TRPA1 mRNA and protein level	Akan et al., 2023 [28]
Carvacrol (i.p. for 3 days before lesion. 10 mg/kg)	Male Wistar Rat; 250–300 g. 6-OHDA (12,5 μg, intrastriatal, unilateral)	↓ Apomorphine induced rotations		SN: ↓ MDA, ↓ NO2-, ↑ Catalase Activity			Baluchnejadmojarad et al., 2014 [29]
Carvacrol (i.p. same day as lesion. 40 mg/kg)	Male C57BL/6 Mouse; 3 months old. 6-OHDA (6 μg, intrastriatal, unilateral)	↓ Ipsilateral forelimb use in Cylinder Test	SN: ↑ TH protein CPU: ↑ TH protein			SN: ↓ Caspase 3 protein CPU: ↓ Caspase 3 protein	Dati et al., 2017 [30]
Carvacrol (i.p. for 49 days [7 before and 42 after lesion]. 25 mg/kg, 50 mg/kg, 100 mg/kg)	Male Wistar Rat; 250–350 g. 6-OHDA (16 μg, intra-MFB, unilateral)	≈ Apomorphine induced rotations		CPU: ≈ MDA, ≈ Thiol		↑ Performance in Passive Avoidance Test ≈ Tail Flick Latency	Haddadi et al., 2018 [31]
Carvacrol (i.p. for 49 days [7 before and 42 after lesion]. 25 mg/kg)	Male Wistar Rat; 250–300 g. 6-OHDA (16 μg, intra-MFB, unilateral)	≈ Apomorphine induced rotations (↓ with physical exercise)		CPU: ≈ MDA (↓ with physical exercise), ≈ Thiol (↑ with physical exercise) Hipp: ≈ MDA (↓ with physical exercise), ≈ Thiol		↑ Performance in Passive Avoidance Test	Hamzehloei et al., 2019 [32]
Carvacrol (i.p. for 30 days along with lesion. 12.5 mg/kg, 25 mg/kg)	Male Wistar Rat; 7 months old. Reserpine s.c. 0.1 mg/kg for 30 days every other day	↓ Catalepsy ≈ Distance traveled in Open Field ↓ Vacuous Chewing Movements ≈ Rearing in Open Field	SN: ↑ TH positive cells CPU: ↑ TH optic density				Lins et al., 2018 [33]
Carvacrol (i.p. for 15 days after lesion. 10 mg/kg, 15 mg/kg, 20 mg/kg)	Male Wistar Rat; 200–250 g. 6-OHDA (16 μg, intrastriatal, unilateral)	↓ Apomorphine induced rotations ↓ Catalepsy ↓ Inversion and Total time in Pole Test ↓ Inversion and Total time in Beam Walking Test ↑ Time to fall from Rotarod ↑ Distance traveled in Rotarod ≈ Distance traveled in Open Field		CPU: ↑ GSH, ↓ MDA			Manouchehrabadi et al., 2020 [34]
Carvacrol (p.o. for 15 days before lesion. 25 μg/kg)	Male Wistar Rat; 60 days old. 6-OHDA (10 μg, intra-SN, unilateral)		SN: ↑ TH positive signal		↓ CSF IL-1β ↓ Serum IL-1β ↓ Serum TNF-α	Normal blood cell count and serum aminotransferases activity	Ribeiro et al., 2019 [35]
Carvacrol (p.o. for 21 days along with lesion. 25 mg/kg, 50 mg/kg, 100 mg/kg)	Male Swiss Mouse; 10–12 weeks old. Rotenone i.p. 1.5 mg/kg for 21 days.	↑ Time to fall from Rotarod ↑ Distance traveled and maximum speed in Open Field ↓ Freezing time and immobility time in Open Field	SN: ↑ TH positive signal and protein	SN: ↑ NRF2 protein, ↑ NQO-1 protein, ↑ HO-1 protein	SN: ↓ NLRP3 protein, ↓ ASC protein, ↓ TNF-α, ≈ IL-6, ↓ IL-1β, ↑ IL-4	↓ α-synuclein in SN	Shah et al., 2024 [36]
Citronellol (p.o. for 28 days along with lesion. 25 mg/kg)	Male Wistar Rat; 280–300 g. Rotenone i.p. 2.5 mg/kg for 28 days.		SN: ↑ TH positive cells CPU: ↑ TH positive fibers	SN: ↑ Catalase, ↑ GSH, ↑ Nrf2, ↓ MDA, ↑ SOD	SN: ↓ IL-1β, ↓ IL-6, ↓ MMP-9, ↓ TNF-α, ≈ COX-2, ↓ iNOS CPU: ↓ IBA1, ↓ GFAP	↓ α-synuclein in SN ↓ Bax, ≈ Bcl-2, ≈ mTOR in SN ↓ LC-3, ≈ p62 in SN	Jayaraj et al., 2022 [37]
Geraniol (p.o. for 7 days before and along with lesion. 100 mg/kg)	Male C57BL/6 Mouse; 25–30 g. MPTP i.p. 30 mg/kg for 4 days.	↑ Time to fall from Rotarod ↑ Number of Steps in Drag Test ↑ Fore Paw Stride Length in Foot Print Test	CPU: ↑ DA, ↑ DOPAC, ↑ HVA, ↑ TH protein, ↑ DAT protein, ↑ VMAT-2 protein SN: ↑ TH positive cells	CPU: ↑ MDA, ↑ GSH		↑ BDNF (protein and mRNA), ↑ GDNF (protein and mRNA) in CPU	Rekha et al., 2013 [38]
Geraniol (p.o. for 7 days along with lesion. 50 mg/kg, 100 mg/kg, 200 mg/kg)	Male C57BL/6 Mouse; 25–30 g. MPTP i.p. 30 mg/kg for 7 days.	↑ Hanging Time in Hang Test	CPU: ↓ MAO-B activity, ↑ TH positive fibers SN: ↑ DA, ↑ DOPAC, ↑ HVA			↓ α-synuclein mRNA and protein in SN and CPU	Rekha et al., 2013 [39]
Geraniol (p.o. 10 doses along with lesion through 35 days. 100 mg/kg)	Male C57BL/6 Mouse; 25–30 g. 10 doses of MPTP 25 mg/kg s.c. with probenecid 250 mg/kg i.p. through 35 days.	↓ Bradykinesia in Pole Test ↑ Distance traveled in Open Field ↓ Catatonia ↑ Rearing in Open Field	SN: ↑ DAT positive cells	CPU: ↑ GSH-Px activity, ↓ SOD activity, ↓ Catalase activity		↑ Grooming in Open Field ↑ Bcl-2 mRNA and protein, ↓ Bax mRNA and protein, ↓ Cyc1 mRNA and protein, ↓ Caspase-9 mRNA and protein in SN ↑ Bcl-2 protein, ↓ Bax protein, ↓ Cyc1 protein and ↓ Caspase-9 protein in CPU	Rekha and Selvakumar, 2014 [40]
Limonene (i.p. 5 days a week for 4 weeks along with lesion. 50 mg/kg)	Male Wistar Rat; 260–300 g. Rotenone i.p. 2.5 mg/kg 5 days a week for 4 weeks.	↑ Time to fall from Rotarod	SN: ↑ TH positive cells CPU: ↑ TH positive fibers	SN: ↓ MDA, ↑ SOD activity, ↑ Catalase, ↑ GSH	SN: ↓ TNF-α, ↓ IL-6, ↓ IL-1β CPU: ↓ GFAP positive cells, ↓ IBA1 positive cells, ↓ iNOS, ↓ COX-2, ↓ p-NFκB, ↓ p-IκB	↓ α-synuclein in CPU ↑ BDNF in CPU ↓ Phosphorylation of P38 and JNK in CPU ↑ Phosphorylation of mTOR in CPU ↑ Mitochondrial Complex 1 in CPU ↓ Bax, ↑ Bcl2, ↓ Cl-Caspase 3, ↓ Cl-Caspase 9, ↓ Cytochrome-C, ↓ CHOP and ↓ p-MST1 in CPU	Eddin et al., 2023 [41]
Linalool (p.o. for 15 days after lesion. 25 mg/kg, 50 mg/kg, 100 mg/kg)	Male Wistar Rat; 250–280 g. 6-OHDA (12 μg, intrastriatal, unilateral)	↓ Apomorphine induced rotations ↑ Distance traveled in Open Field	CPU: ↑ DA, ↑ DOPAC, ≈ HVA, ↑ TH positive fibers, ↑ DAT positive fibers	↓ Nitrites and ↓ MDA in CPU, Hipp, and PFC			de Lucena et al., 2020 [42]
Linalool (p.o. for 7 days before lesion. 12.5 mg/kg, 25 mg/kg).	Male C57BL/6 Mouse; 20–25 g. MPTP 4 i.p. injections. 20 mg/kg	↑ Grip Strength ↑ Latency to Fall in Wire Hang Test ↑ Distance traveled in Open Field ↑ Mean velocity in Open Field	SN: ↑ TH positive area			↑ Time in Central Zone in Open Field ↓ Immobility Time and ↑ Swimming Time in Forced Swimming Test ↑ Time spent in open arms, ↑ Open arm entries and ↓ Time spent in closed arms in Plus-Maze test	Chang et al., 2024 [43]
Menthol (p.o. for 28 days after lesion. 10 mg/kg, 20 mg/kg).	Male Wistar Rat; 280–320 g. LPS (intra-SN, unilateral)	↓ Apomorphine induced rotations	SN: ↑ TH positive cells and protein		SN: ↓ IBA1 positive cells and protein, ↓ iNOS protein and mRNA, ↓ COX-2 protein and mRNA, ↓ IL-1β mRNA, ↓ IL-6 mRNA, ↓ TNF-α mRNA		Du et al., 2020 [44]
Myrcene (p.o. 5 days a week for 4 weeks along with lesion. 50 mg/kg)	Male Wistar Rat; 280–300 g. Rotenone i.p. 2.5 mg/kg 5 days a week for 4 weeks.		SN: ↑ TH positive cells CPU: ↑ TH positive fibers	SN: ↓ MDA, ↑ GSH, ↑ Catalase, ↑ SOD activity	SN: ↓ TNF-α, ↓ IL-6, ↓ IL-1β, ↓ MMP-9, ↓ iNOS, ↓ COX-2 CPU: ↓ IBA1 positive cells, ↓ GFAP positive cells	SN: ↓ Bax, ↑ Bcl-2, ↓ Bax/Bcl-2 ratio CPU: ↓ α-synuclein, ↓ Beclin-1, ↓ LC3B, ↓ P62, ↑ mTOR phosphorylation	Azimullah et al., 2023 [45]
Myrtenal (i.p. for 5 days before lesion. 50 mg/kg)	Male Wistar Rat; 250–300 g. 6-OHDA (10 μg, intrastriatal, unilateral)	↓ Apomorphine induced rotations ↓ Number of falls in Rotarod	↑ DA in whole brain	↓ MDA in ipsilateral side of brain, ≈ GSH, ≈ SOD, ↑ Catalase in whole brain, ≈ GSH-Px		Slight body weight loss after pre-treatment. ↑ Step-through Latency in Passive Avoidance Test	Tancheva et al., 2020 [46]
Myrtenol (p.o. for 28 days along with lesion. 5 mg/kg)	Male Swiss Mouse; 45–65 g. Reserpine s.c. 0.1 mg/kg every two days for 28 days	Delayed catalepsy ↓ Vacuous Chewing Movements ≈ Distance traveled and average speed in Open Field	≈ TH+ positive cells in SN and VTA ↑ TH+ positive fibers in CPU	PFC: ↓ MDA, ≈ Total Antioxidant Status, Total Oxidant Status and Oxidative Stress Index Hipp: ≈ MDA, Total Antioxidant Status, Total Oxidant Status and Oxidative Stress Index CPU: ≈ MDA and Total Antioxidant Status, ↓ Total Oxidant Status and Oxidative Stress Index		↑ Olfactory discrimination ↑ Novel Object Recognition ≈ Time in Open Arms in Plus-Maze Test ≈ Central Zone Entries in Open Field	Silva-Martins et al., 2021 [47]
Perillyl Alcohol (p.o. for 14 days [7 before and 7 after lesion]. 100 mg/kg)	Male Wistar Rat; 250–300 g. 6-OHDA (16 μg, intrastriatal, unilateral)	↓ Motor Asymmetry ↓ Escape Latency in Narrow Beam Test ≈ Time to Cross in Narrow Beam Test ↑ Rearing	CPU: ↑ TH mRNA and protein	CPU: ↓ Intracellular ROS Generation, ↑ Nrf2 protein	CPU: ↓ IL-1β mRNA, ↓ TNF-α mRNA	CPU: ↓ DNA Fragmentation, ↓ Caspase 3 Activation, ↓ Bax mRNA, ↑ Bax protein, ↑ Bcl-2 mRNA and protein, ↑ PGC-1α mRNA and protein, ↑ Drp1 protein	Anis et al., 2020 [48]
α-pinene (nanoemulsion p.o. for 6 days after lesion). 50 mg/kg, 100 mg/kg.	Male Wistar Rat; 200–220 g. Reserpine 5 mg/kg i.p. single dose.	↓ Tremulous Jaw Movements ↑ Rearing ↓ Catalepsy ↓ Hind Limb Rigidity		Whole Brain: ↓ MDA, ↑ SOD Activity, ↑ Catalase Activity, ↑ Glutathione, ↓ NO Level		↑ Grooming Behavior	Srivastava et al., 2021 [49]
α-pinene (nanoemulsion p.o. for 15 days before lesion). 50 mg/kg, 100 mg/kg.	Male Wistar Rat; 200–220 g. Haloperidol 1 mg/kg i.p. single dose	↓ Catalepsy ↑ Swimming Performance ↑ Locomotor Performance ↓ Akinesia ↑ Time to fall from Rotarod					Srivastava et al., 2021 [49]
α-pinene (nanoemulsion p.o. for 28 days along with lesion [7 days a week]). 50 mg/kg, 100 mg/kg)	Male Wistar Rat; 200–220 g. Rotenone 1.5 mg/kg s.c. for 28 days.	↑ Time to fall from Rotarod ↑ Time to fall from Hanging Wire ↑ Locomotor Activity ↑ Postural Stability ↑ Step Alternation ↑ Forelimb Locomotion		Whole Brain: ↓ MDA, ↑ SOD Activity, ↑ Catalase Activity, ↑ Glutathione, ↓ NO Level			Srivastava et al., 2024 [50]
α-pinene (nanoemulsion p.o. for 28 days along with lesion [5 days a week in the last 28 days]). 50 mg/kg, 100 mg/kg)	Male Wistar Rat; 200–220 g. Trichloroethylene 1000 mg/kg p.o. for 8 weeks (5 days a week).	↑ Time to fall from Rotarod ↑ Time to fall from Hanging Wire ↑ Locomotor Activity ↑ Postural Stability ↑ Step Alternation ↑ Forelimb Locomotion		Whole Brain: ↓ MDA, ↑ SOD Activity, ↑ Catalase Activity, ↑ Glutathione, ↓ NO Level			Srivastava et al., 2024 [50]
Safranal (i.p. for 14 days. 0.1 mg/kg, 0.2 mg/kg, 0.4 mg/kg)	Male C57BL/6 Mouse. MPTP 25 mg/kg i.p. for 5 days.	↓ Catalepsy ↑ Grip Strength ↑ Distance traveled in Open Field ↑ Performance in Pole Climbing ↑ Time to fall from Rotarod ↑ Swing speed and stride length of hindlimbs and forelimbs	CPU: ↑ TH protein, mRNA and fluorescence intensity, ↑ DA		SN: ↓ NLRP3 mRNA and protein, ↓ IL-1β mRNA and protein	SN: ↓ Caspase-1 mRNA and protein	Yang et al., 2024 [51]
Thymol (i.p. for 28 days along with lesion. 50 mg/kg)	Male Wistar Rat; 280–300 g. Rotenone 2.5 mg/kg i.p. for 28 days.		SN: ↑ TH positive cells CPU: ↑ TH positive fibers	SN: ↓ MDA, ↑ GSH, ↑ SOD Activity, ↑ Catalase Activity	CPU: ↓ GFAP positive cells, ↓ Iba-1 positive cells, ↓ COX-2 protein, ↓ iNOS protein SN: ↓ IL-1β, ↓ IL-6, ↓ TNF-α		Javed et al., 2019 [52]
Thymol (i.p. for 15 days after lesion. 20 mg/kg, 30 mg/kg, 40 mg/kg)	Male Wistar Rat; 200–250 g. 6-OHDA (15 μg, intrastriatal, unilateral)	↓ Apomorphine induced rotations ↓ Catalepsy ↓ Inversion and Total time in Pole Test ↓ Inversion and Total time in Beam Walking Test ≈ Time to fall from Rotarod ↑ Distance traveled in Rotarod ↑ Distance traveled in Open Field		CPU: ↑ GSH, ↓ MDA			Nourmohammadi et al., 2022 [53]
Thymol (p.o. for 35 days along with lesion. 30 m/kg)	Male Sprague-Dawley Rat; 300–320 g. Mn2+ 10 mg/kg i.p. for 35 days	↓ Latency to move in Open Field ↑ Distance traveled in Open Field ↑ Rearing in Open Field ↓ Latency to move in Swimming Test ↓ Swimming Time ↑ Swimming Direction Score ↓ Catalepsy	↑ DA in whole brain	↑ Nrf2 in whole brain ↑ HO-1 in whole brain ↑ Total Antioxidant Capacity in whole brain ↑ SOD in whole brain ↓ MDA in whole brain	↓ COX-2 in whole brain ↓ iNOS in whole brain ↓ TNF-α in whole brain ↓ TLR4 in whole brain ↓ NLRP3 in whole brain ↓ NFκB in whole brain ↓ IL-1β in whole brain ↓ GFAP mRNA in whole brain ↓ Aif mRNA in whole brain	↑ Grooming in Open Field ↑ Spontaneous Alternation in Y-Maze Test ↑ NE, 5-HT and GABA in whole brain ↓ ACHE activity in whole brain ↑ BDNF in whole brain ↓ Glutamate in whole brain ↓ Caspase 1, Caspase 3 and BAX mRNA in whole brain ↑ Bcl2 mRNA in whole brain ↓ GSK-3β mRNA in whole brain ≈ Histopathological alterations in cerebral cortex ↓ Histopathological alterations in Hipp and CPU	Abu-Elfotuh et al., 2022 [54]
Thymoquinone (p.o. for 2 days before lesion. 5 mg/kg, 10 mg/kg)	Male Wistar Rat; 190–250 g. 6-OHDA (12.5 μg, intrastriatal, unilateral)	↓ Apomorphine induced rotations		SN: ↓ MDA, ≈ Nitrites, ≈ SOD		↑ Nissl stained neurons in SN	Sedaghat et al., 2014 [55]
Thymoquinone (p.o. every 2 days for 30 days along with lesion. 7.5 mg/kg, 15 mg/kg)	Male Wistar Rat; 250–350 g. Rotenone 1 mg/kg s.c. every 2 days for 30 days.	≈ Time to fall from Rotarod ≈ Rearing in Open Field ≈ Catalepsy	SN: ↑ DA, ↑ TH positive cells CPU: ↑ TH positive fibers	↓ Serum pro-oxidant/antioxidant balance		≈ Body Weight ≈ Parkin in SN and CPU ↓ Drp1 in SN and CPU	Ebrahimi et al., 2017 [56]
Thymoquinone (i.p. for 1 week along with lesion. 10 mg/kg)	Male C57BL/6 Mouse; 25–30 g. MPTP 25 mg/kg i.p. for 5 days		SN: ↑ TH positive cells CPU: ↑ DAT positive fibers	CPU: ↑ SOD Activity, ↑ Catalase Activity, ↓ MDA, ↑ GSH	CPU: ↓ IL-1β, ↓ IL-6, ↓ TNF-α, ↓ COX-2 protein, ↓ iNOS protein		Ardah et al., 2019 [57]
Thymoquinone (i.p. for 7 days before and along with lesion. 10 mg/kg)	C57BL/6 Mouse; 25–30 g. MPTP 25 mg/kg i.p. for 5 days		SN: ↑ TH positive cells	SN: ↓ MDA, ↑ SOD Activity, ↑ GSH-Px Activity, ↓ 8OH-dG, ↑ Nrf2, ↑ HO-1, ↑ NQO-1, ↑ GST		SN: ↓ α-synuclein, ↓ Fluoro Jade staining	Dong et al., 2021 [58]

### 2.4. Effects of MTS in Motor Behavioral Parameters

Motor symptoms in PD, being the most visible and evident, allow for a rapid assessment of dopaminergic neurodegeneration in the SN and the reduction of cerebral dopamine levels. For this reason, the most commonly employed behavioral tests were Induced Rotation (in 6-OHDA models), Open Field, Rotarod, and Catalepsy tests. Elevated Body Sway, akinesia, and swimming movements were employed in minor cases, being reported in only one work each [48,49,54]. Most reports demonstrated improvements in evaluated motor parameters following MTs’ treatment (Table 1).

With respect to the Rotarod test, which evaluates motor coordination through locomotor activity on a rotating rod [59], 8 out of 10 publications reported that PDAs took significantly more time to fall off the apparatus when administered with at least one of the doses evaluated in MTs, mostly the higher dose (20–50 mg/kg i.p. and 50–100 mg/kg p.o.). Tancheva et al. [46] measured the number of falls and reported less falls in PDAs administered with myrtenal than PDAs. A study by Nourmohammadi et al. [53] on thymol reported that it increased the time of stay on the Rotarod in animals injected with 6-OHDA, though not significantly; however, it was capable of reversing the negative effect on the distance traveled in the rod. The meta-analysis of time-to-fall measurements using the Rotarod apparatus revealed a significant decrease in the duration that PDAs remained on the rotating rod compared to control animals (SMD = −6.05 [95% CI: −8.45 to −3.65] *p* < 0.01 I^2^ = 93%) (Appendix A). PDAs with monoterpene therapy (PDAs + MTs) remained on the Rotarod longer before falling compared to PDAs (SMD = 2.44 [95% CI: 1.77 to 3.11] *p* < 0.01 I^2^ = 87%) (Figure 3A). This suggests that the administration of MTs enhances motor coordination in PDAs. However, compared to control animals, PDAs + MTs took less time to fall (SMD = −3.23 [95% CI: −4.14 to −2.33] *p* < 0.01 I^2^ = 91%) (Figure 3B).

Another important behavioral test analyzed was the Catalepsy test, which assesses postural rigidity and ability to start movements [60]. Seven out of nine publications analyzed reported that PDAs + MTs took less time to start a movement than PDAs. Meta-analysis of the latency of movement showed that PDAs took more time to finish the test than control animals (SMD = 5.31 [95% CI: 3.00 to 7.63] *p* = 0.00001 I^2^ = 89%) (Appendix A). The PDAs + MTs showed to decrease the latency in the Catalepsy test, evidencing an attenuation of the motor impairment induced in these animals (SMD = −2.63 [95% CI: −3.71 to −1.54] *p* < 0.01 I^2^ = 88%) (Figure 4A). The latency of the PDAs + MTs was not as low as the latency of control animals, indicating that they still showed more rigidity than healthy subjects (SMD = 3.66 [95% CI: 2.51 to 4.80] *p* < 0.01 I^2^ = 85%) (Figure 4B).

The high heterogeneity (I^2^ ≥ 80%) observed in the meta-analysis of behavioral test results is likely due to the use of different experimental protocols and animal models across studies. Variations in treatment dosages, time points, and animal strains can contribute to this variability, as well as differences in the set-up of the behavioral test.

We decided to exclude induced rotation tests from the meta-analysis, since it could be only measured in models that administered a neurotoxic compound unilaterally in the nigrostriatal circuit. The Open Field test was also excluded because of extensive differences in protocols, units of measurement, and parameters measured.

### 2.5. Effect of MTs on Dopamine-Related Parameters

PDAs show decrease in DA content, DA metabolites, and DA neuron markers, such as TH, DAT, and VMAT, in SN and CPU. Publications with DAT and/or VMAT measurements reported an increase in their content with MT therapy, particularly, borneol, geraniol, linalool, and thymoquinone [27,38,40,42,57]. Other studies measure DA in brain tissue with chromatographic or fluorometric techniques, showing an increase of DA concentration, in SN, CPU, or whole brain, in PDAs-MTs [27,38,39,42,46,51,54,56].

Meta-analysis of the results of TH shows a decrease in PDAs compared to control animals. This was analyzed both in the SN, where the nuclei of dopaminergic neurons are located (SMD = −3.85 [95% CI: −5.02 to −2.68] *p* < 0.01 I^2^ = 64%), and in the CPU where these neurons project (SMD = −4.78 [95% CI: −6.61 to −2.94] *p* < 0.01 I^2^ = 69%) (Appendix A). Among the 20 publications included in the study, 10 measured TH in both the SN and the CPU, 6 focused exclusively on the SN, and 4 concentrated solely on the CPU. The most reported monoterpene was carvacrol, appearing in 4 studies that measured TH. Results from PDAs showed that PDAs + MTs had a significantly higher TH content, both in the SN (SMD = 3.02 [95% CI: 2.14 to 3.90] *p* < 0.01 I^2^ = 67%) and in CPU (SMD = 2.89 [95% CI: 1.94 to 3.84] *p* < 0.01 I^2^ = 64%) (Figure 5A). The effect of MTs in increasing TH levels was similar in both brain regions, though it was slightly greater in SN. Differences between doses evaluated in a single study were varied, although most studies reported that TH content increased the most with the highest dose. Ebrahimi et al. [56] reported an increase of the TH-enhancing effect with 15 mg/kg thymoquinone in CPU, compared with 7.5 mg/kg thymoquinone, but a decrease in SN. Lins et al. [33] reported that a dose of 25 mg/kg of carvacrol led to an increase in TH levels in both brain structures compared to 12.5 mg/kg. Rekha et al. [39] reported that 100 mg/kg geraniol was able to increase TH signal compared with 50 mg/kg, but also reported that the effect with 200 mg/kg geraniol was lower. Still, meta-analysis showed that PDAs + MTs had significantly lower TH content in SN (SMD = −2.17 [95% CI: −2.91 to −1.42] *p* < 0.01 I^2^ = 66%) and CPU (SMD = −2.59 [95% CI: −3.64 to −1.55] *p* < 0.01 I^2^ = 74%) than control animals (Figure 5B).

Heterogeneity in the TH meta-analyses was moderate (50% ≤ I^2^ < 80%), but lower than in the behavioral test meta-analyses. This reflects the existence of variability in TH experimental protocols, albeit lower than the differences in animal models and therapy administration.

All the results presented here suggest that MTs may halt or delay dopaminergic neuron death in these animal models, although they do not fully restore the neuron numbers observed in the control animals. Figure 6 summarizes the effects of MTs on dopamine-related parameters.

### 2.6. Antioxidant Properties of MTs

Among the techniques used in the publications analyzed, the main one was the quantification of malondialdehyde (MDA), a byproduct of lipid peroxidation, through thiobarbituric acid reactive substances (TBARS) reaction [61], followed by evaluation of the superoxide dismutase (SOD), catalase (CAT), and glutathione peroxidase (GPx) activity. Measurement of reduced glutathione (GSH), nitrites, and reactive oxygen species (ROS) has also been performed to quantify oxidative stress. Other markers include 8-OHdG, a marker for oxidation of RNA [58], and components of the NRF2 antioxidant pathway (Table 1).

Meta-analysis of results from MDA quantification assays shows that PDAs present higher MDA content than control animals both in CPU (SMD = 2.21 [95% CI: 0.39 to 4.03] *p* < 0.01 I^2^ = 88%) and SN (SMD = 2.91 [95% CI: 2.02 to 3.81] *p* < 0.01 I^2^ = 50%) (Appendix A). However, it is observed that the PDAs + MTs exhibit lower levels of MDA and, consequently, reduced lipid peroxidation compared to the PDAs in both the CPU (SMD = −2.69 [95% CI: −3.82 to −1.57] *p* < 0.01 I^2^ = 87%) and SN (SMD = −2.62 [95% CI: −3.57 to −1.66] *p* < 0.01 I^2^ = 69%) (Figure 7A). When compared with control animals, PDAs + MTs still had higher MDA content in both structures (CPU: SMD = 1.21 [95% CI: 0.34 to 2.07] *p* < 0.01 I^2^ = 85%; SN: SMD = 0.95 [95% CI: 0.57 to 1.33] *p* < 0.01 I^2^ = 4%) (Figure 7B).

Determination of antioxidant enzymes’ activity or concentration in biological samples is another useful parameter to measure the oxidative state of the brain. CAT and SOD are the two main antioxidant enzymes reported in the literature. Meta-analysis of results shows that PDAs have reduced the activity and/or content of the antioxidant enzymes in the SN (CAT: SMD = −2.68 [95% CI: −3.82 to −1.54] *p* < 0.01 I^2^ = 59%; SOD: SMD = −3.64 [95% CI: −5.05 to −2.23] *p* < 0.01 I^2^ = 70%) (Appendix A). On the other hand, PDAs + MTs achieve higher antioxidant enzyme content and/or activity than PDAs according to the meta-analysis results (CAT: SMD = 1.74 [95% CI: 0.90 to 2.57] *p* < 0.01 I^2^ = 49%; SOD: SMD = 2.44 [95% CI: 1.24 to 3.65] *p* < 0.01 I^2^ = 80%) (Figure 8A). Compared to control animals, however, concentration/activity of these enzymes remained significantly lower at the time of measurement (CAT: SMD = −0.78 [95% CI: −1.27 to −0.28] *p* < 0.01 I^2^ = 0%; SOD: SMD = −0.93 [95% CI: −1.34 to −0.53] *p* < 0.01 I^2^ = 6%) (Figure 8B).

### 2.7. Anti-Inflammatory Properties

Enzyme-Linked ImmunoSorbent Assay (ELISA) and Western Blot were the primary techniques used in the analyzed publications to quantify the protein content of inflammatory mediators, such as cytokines and inflammasome-associated proteins. Quantitative polymerase chain reaction (qPCR) to measure mRNA content was performed in several studies to complement the protein measurements or as individual studies. Glial cell quantification was conducted using ionized calcium-binding adaptor molecule 1 (IBA-1) and/or glial fibrillary acidic protein (GFAP) immunostaining, which serve as markers for microglia and astrocytes, respectively. Most studies link glial activation to a pro-inflammatory environment, and the publications reviewed here indicate that MTs reduce the immunostaining signal for both microglia and astrocytes (Table 1).

The main inflammatory parameters assessed were the concentration or gene expression levels of pro-inflammatory cytokines, with most publications that measured them (62%) focusing on the SN. The meta-analysis of results from quantification of cytokines IL-1, TNF-α, and IL-6 in SN, shows their significant increase in PDAs compared to control animals (IL-1: SMD = 7.21 [95% CI: 4.36 to 10.05] *p* < 0.01 I^2^ = 81%; IL-6: SMD = 4.14 [95% CI: 2.01 to 6.27] *p* < 0.01 I^2^ = 77%; TNF-α: SMD = 6.20 [95% CI: 2.86 to 9.54] *p* < 0.01 I^2^ = 86%) (Appendix A). PDAs + MTs showed lower concentration or gene expression of pro-inflammatory cytokines in SN than PDAs (IL-1: SMD = −3.79 [95% CI: −5.72 to −1.85] *p* < 0.01 I^2^ = 89%; IL−6: SMD = −3.37 [95% CI: −4.77 to −1.98] *p* < 0.01 I^2^ = 76%; TNF-α: SMD = −6.46 [95% CI: −8.89 to −4.02] *p* < 0.01 I^2^ = 82%) (Figure 9A). Standardized mean differences for the TNF-α were lower than for IL-1 and IL-6. When compared with control animals, IL-1 and IL-6 remained higher in concentration/gene expression levels in PDAs + MTs animals. However, TNF-α levels in PDAs + MTs were not significantly different from values in control animals (IL-1: SMD = 3.70 [95% CI: 2.29 to 5.11] *p* < 0.01 I^2^ = 83%; IL-6: SMD = 2.46 [95% CI: 0.98 to 3.94] *p* < 0.01 I^2^ = 84%; TNF-α: SMD = 1.76 [95% CI: −0.20 to 3.73] *p* = 0.078 I^2^ = 87%) (Figure 9B). This result suggests that MTs restore the levels of a key pro-inflammatory cytokine to control values in animals with parkinsonian pathology.

### 2.8. Limitations Section

It is important to note that the considerable variability in animal models, parkinsonism induction protocols, measurement methods across studies, and the diverse types of MTs with varying functional groups and structures may have contributed to the high heterogeneity observed in many of the analyzed results (I^2^ ≥ 80%). Therefore, a more in-depth analysis of the MTs subgroups is required. This will be achievable in the future when enough studies are available to perform a robust meta-analysis. Another limitation was the number of publications that were excluded from the meta-analysis because of discrepancies in the results and those that were not available for analysis, which could render the data sets extracted incomplete. Moreover, some included studies have high or moderate risk of bias, which might have affected the outcomes. The most common concern was selection bias, as many studies did not report clear randomization criteria for assigning individuals to experimental groups. Additionally, performance and detection bias were frequently identified due to a lack of information on whether experiments were conducted and analyzed under blinded conditions. The most significant risk of bias was observed in relation to incomplete outcome reporting. Many studies failed to provide detailed sample sizes (n) for each experimental group, did not account for attrition or exclusions in their analyses (attrition bias), or exhibited inconsistencies between the numerical results described in the text and those shown in the figures (other bias). In contrast, reporting bias was the least concerning issue, as most studies adequately described their statistical analyses and provided complete references for their results.

## 3. Discussion

PD is a complex condition that affects the quality of life of people who suffer from it, mainly aged individuals. Current pharmacological therapies, as well as surgical treatments and supportive therapies, have had modest efficacy in reducing motor symptoms [62]. Plant-derived active compounds, primarily secondary metabolites like EOs, are known to modulate various mechanisms involved in pathological processes. Due to their pleiotropic effects, these compounds offer broader and more effective neuroprotection compared to single-target agents. MTs, the main components of EOs, have been historically used to treat a range of pathological conditions [24,63]. MTs demonstrated to improve neurotoxin-induced motor deficits (PD’s hallmark symptoms in its late stages) in different PD animal models through a variety of behavioral tests, primarily the Rotarod and Catalepsy tests. In the Rotarod test, most studies reported that PDAs + MTs spent significantly more time in the accelerating spinning rod than PDAs (Figure 3), typically with doses between 25 and 100 mg/kg. However, certain MTs enhance motor performance at significantly lower concentrations. For example, safranal, a cyclic oxygenated monoterpene with an aldehyde functional group, has shown effects at doses as low as 0.1 mg/kg [51]. Meta-analysis of the Catalepsy test results showed that PDAs + MTs took significantly less time to lower their paws from the elevated bar than PDAs (Figure 4). Although the meta-analysis of both tests indicated that phytotherapy with MTs improved motor coordination and reduced rigidity in PDAs, MTs did not fully counteract the effects of Parkinson-inducing toxins, since in these parameters evaluated, no significant differences were observed with respect to control animals.

Other behavioral processes, besides motor ones, have been described in many publications. Sensorial function was evaluated by Silva-Martins et al. [47] who showed that myrtenol administration mitigated significantly the negative effects of reserpine in olfactory discrimination. However, Haddadi et al. [31] did not find significant differences in tail-flick latency between PDAs and PDAs treated with carvacrol. MTs also demonstrated a potential mitigating effect on anxiety-related manifestations observed in PDAs. Geraniol, α-pinene, and thymol caused an increase in grooming behavior compared to PDAs [40,49,54], while borneol caused a decrease in freezing behavior in PDAs + MTs [27]. Chang et al. [43] reported that PDAs administered with linalool showed more time in the center area of the Open Field test, less immobility time in the forced-swimming test, and more time in the open arms of the Plus-Maze test. Cognitive processes such as memory tasks were also explored by performing the passive avoidance test with carvacrol [31,32] and myrtenal therapy [46]. All studies showed a better performance of PDAs + MTs compared to PDAs. Silva-Martins et al. [47] showed that myrtenol enhanced novel object recognition in PDAs, while Abu-Elfotuh et al. [54] found that thymol enhanced memory in PDAs measured as more spontaneous alternations in the Y-Maze test. These results demonstrated that MTs can also have anxiolytic, antidepressant, memory-enhancing, and sensorial effects, likely through the modulation of glutamate and gamma-aminobutyric acid neurotransmitter systems, as reported by Agatonovic-Kustrin et al. [64]. This further supports their therapeutic potential for the treatment of non-motor symptoms of PD.

Since the loss of dopaminergic neurons in the SN characterizes the disease, the expression and activity of TH, DAT, and VMAT proteins—key components of the nigrostriatal dopaminergic pathway—are significantly affected. MTs have shown to increase DA levels in various psychiatric disorders [65]. Moreover, the meta-analysis of TH immunostaining performed here demonstrated that MTs significantly increase the number of TH-positive neurons, which synthesize DA, in the SN and CPU of PDAs (Figure 5A). The mechanism of MTs remains unclear, though it is believed that they may protect dopaminergic neurons from oxidative damage. The literature suggests that the antioxidant and anti-inflammatory properties of MTs may contribute to neuronal protection, and this hypothesis is further supported by all the meta-analysis performed in the present work. Plenty of evidence supports the antioxidant properties of MTs like limonene, myrtenol, and α-pinene [66,67,68], typically assessed by measuring lipid (MDA) or protein oxidation and antioxidant enzyme activity (CAT, SOD). Studies of their antioxidant mechanisms reveal that MTs inhibit pro-oxidant enzymes (NADPH oxidases, cyclooxygenases), enhance cellular antioxidant components, and neutralize free radicals [69,70]. However, the effect of MTs on cellular oxidative status may depend on factors such as dosage, biological context, and experimental conditions. Some studies have reported that MTs can either reduce or elevate MDA levels in tissues, highlighting their context-dependent influence on oxidative balance [71,72,73,74,75]. Despite these discrepancies, a meta-analysis of MDA quantification indicates that PDAs + MTs, at doses ranging from 10 mg/kg to 100 mg/kg, consistently resulted in significantly lower MDA concentrations in the SN and CPU compared to PDAs (Figure 7A). On the other hand, meta-analysis of SOD and CAT activity showed higher enzymatic activity in the SN of PDAs + MTs compared to PDAs. All these results show that MTs decrease oxidative stress in PDAs, suggesting potential benefits toward treating diseases characterized by oxidative stress, such as atherosclerosis, diabetes, cancer, and neurological diseases [76,77,78,79,80]. The anti-inflammatory effects of MTs, demonstrated in various studies [11], are primarily attributed to their modulation of cytokine balance and production through the NF-κB pathway, leading to a reduction in pro-inflammatory cytokines such as IL-1β, IL-6, and TNF-α [81]. Since inflammation is a known promoter of neurodegeneration, the anti-inflammatory effects of MTs prove to have therapeutic value to treat these conditions. The meta-analysis of results from quantifications of cytokines in the SN shows that the increased levels of IL-1β, IL-6, and TNF-α in PDAs compared to control animals (Appendix A) decreased in PDAs + MTs (Figure 9A).

It is noteworthy that MTs were unable to restore the levels of TH (Figure 5B), MDA (Figure 7B), and the cytokines IL-1β and IL-6 (Figure 9B) to those observed in control animals. However, TNF-α quantification showed that its concentration in the PDAs + MTs was not significantly different from that of the control group (Figure 9B), suggesting a reversal of neurotoxin-induced parkinsonian pathology due to MTs therapy.

All the meta-analyses conducted in this study demonstrated statistically significant evidence supporting the anti-inflammatory and antioxidant effects of 15 MTs in the SN, along with improvements in motor performance and/or dopamine content. These findings substantiate the hypothesis that these mechanisms may act as neuroprotective actions of MTs. Moreover, most studies used oxygenated monoterpenes (e.g., carvacrol, thymol, geraniol, and linalool), suggesting this fraction may drive EOs’ antioxidant and anti-inflammatory effects, as reported by Ainseba et al. [82]. Another mechanism of action proposed in the literature for MTs is the inhibition of apoptosis. Although not included in the meta-analysis due to the limited number of measurements, numerous studies have reported the effects of MTs administration on apoptotic markers in brain tissue [30,37,40,41,45,48,51,54] and mitochondrial integrity, which is closely linked to apoptosis, in brain cells. Most neurotoxic compounds used to induce parkinsonian symptoms in animal models target mitochondria, causing instability and increasing oxidant species production, which drive the neurodegenerative process. Mitochondrial markers assessed in multiple studies indicate that MT administration mitigates mitochondrial dysfunction in PD models [40,41,48,56]. Also, in some murine models, measurements of α-synuclein, the main protein in the bodies of Lewy that are observed in PD post-mortem brains, show that it decreases in PDAs + MTs [36,37,38,41,45,58].

Although MTs comprise a diverse group of compounds with distinct chemical compositions, they share certain physicochemical characteristics that contribute to their similar biological properties. Therefore, all MTs included in this review were analyzed collectively to provide a comprehensive evaluation of their overall effects.

This systematic review demonstrates the potential efficacy of MTs, as bioactive metabolites from plants, in improving motor functions in toxin-based rodent models of PD measured by different behavior tests.

The meta-analyses carried out here quantitatively demonstrate, due to its statistical analysis, that all bibliographic results indicate a neuroprotective activity of MTs, attenuating motor and non-motor manifestations, decreasing brain dopamine content, oxidative stress, and neuroinflammation. However, this does not guarantee successful translation clinically. Although their widespread application in the pharmaceutical and cosmetic industries supports their safety, there are reports of toxicity associated with high-dose ingestion. Low concentrations of MTs, safe for long-term use, could be combined with other treatments to enhance their effects and reduce required pharmaceutical doses.

## 4. Materials and Methods

### 4.1. Search Strategy

The search was focused on control experimental studies of experimental models of PD (rats or mice) that analyzed number or percentage of dopaminergic neurons in relevant structures (SN pars compacta or CPu (CPU)) and/or motor behavior, with mandatory control groups for the parkinsonian model and for the intervention to compare them.

Systematic searches were carried out in PubMed, Scopus, Google Scholar, and Science Direct, regardless of publication date. A first search was conducted between 14 January 2024 and 21 January 2024, and a follow-up search was conducted between 9 December 2024 and 13 December 2024. Details on the search strategy and the study selection criteria are described in the Appendix A. If a publication found in these databases was not freely available, the authors from said publication were contacted to ask for data, if possible. If a negative response was received or there was no response in a period of 2 weeks, the publication was not included in the analysis. During a first screening, the titles and abstracts were evaluated, and in a second screening the content of the remaining articles were assessed.

This study was conducted in accordance with the Preferred Reporting Items for Systematic Reviews and Meta-Analyses (PRISMA) guidelines, with background information on the PRISMA methodology provided in the Appendix A.

The protocol for the meta-analysis was registered in the International Prospective Register of Systematic Reviews (PROSPERO) CRD42024592555 (available from https://www.crd.york.ac.uk/prospero/display_record.php?ID=CRD42024592555, accessed on 6 November 2024). The PICO was constructed as follows: P (population): Animal models of Parkinson’s disease (rats or mice); I (intervention): Monoterpene administration (any formulation, dose and route of administration); C (control): Vehicle administration or no treatment; O (outcome): Motor impairments and changes in biological markers (TH-immunoreactivity, oxidative stress markers, neuroinflammation markers, etc.).

### 4.2. Data Extraction and Evaluation of Risk of Bias

Data and information extraction was performed by one author (MJC), with verifications from a second author (RC). The information extracted included characteristics of the animal (species, strain, sex, weight, and age), the model used to induce parkinsonism (neurotoxin, dose, route and frequency of administration, and control), and the monoterpene therapy (monoterpene, dose, route and frequency of administration, and control). WebPlotDigitizer version 5.2 was used to extract experimental data from graphs. All data were summarized in the systematic review. Data from parameters measured in at least 5 independent publications were included in the meta-analysis.

Risk of bias assessment was performed by two independent reviewers using the Systematic Review Center for Laboratory Animal Experimentation (SYRCLE) tool [83]. Briefly, publications were evaluated according to the sources of bias summarized in the six classifications of the SYRCLE tool (selection bias, performance bias, detection bias, attrition bias, reporting bias, other), with each item categorized as “Low Risk”, “Moderate Risk”, or “Critical Risk” for each article.

### 4.3. Data Analysis

Experimental data were analyzed using the “Metastats” package for R with a random-effects model [84]. The SMD was calculated between PDAs (animals given a Parkinson-inducing neurotoxin) and PDAs + MTs (parkinsonian animals receiving monoterpene therapy). Furthermore, the SMD between the PDAs + MTs group and the control animals group was calculated. All variances in the meta-analysis were analyzed as standard deviation (SD) of the mean; if the variance was expressed as standard error (SE), the SD was calculated according to the following formula: SD = SE×√n. If any measurement was conducted with more than one dose of MTs in a single publication, every dose was analyzed separately. The I^2^ statistic was used to assess heterogeneity, considering a value of I^2^ ≥ 80% high heterogeneity, 50% ≤ I^2^ < 80% moderate heterogeneity, and I^2^ < 50% low heterogeneity [85].

## 5. Conclusions

Phytotherapies with MTs represent a promising therapeutic approach for the prevention and treatment of Parkinson’s disease. Though much research is still needed, main mechanisms of action and evidence of mitigation of symptoms have been reported in multiple studies. Reports of the antioxidant and anti-inflammatory effects of MTs, along with their dopaminergic neuron protection abilities and enhancing of motor performance, provide essential insight on the processes underlying the neurodegenerative process in PD, and highlight the most promising therapeutic targets. Meta-analyses of preclinical studies demonstrated statistically significant improvements in these parameters, providing a quantitative assessment that strengthens current knowledge on the neuroprotective potential of MTs in animal models of PD.

Therefore, this review and its accompanying statistical analysis offer valuable insights into the potential neuroprotective effects of MTs for Parkinson’s disease treatment, contributing to the advancement of knowledge in this field.

## Figures and Tables

**Figure 1 plants-14-00999-f001:**
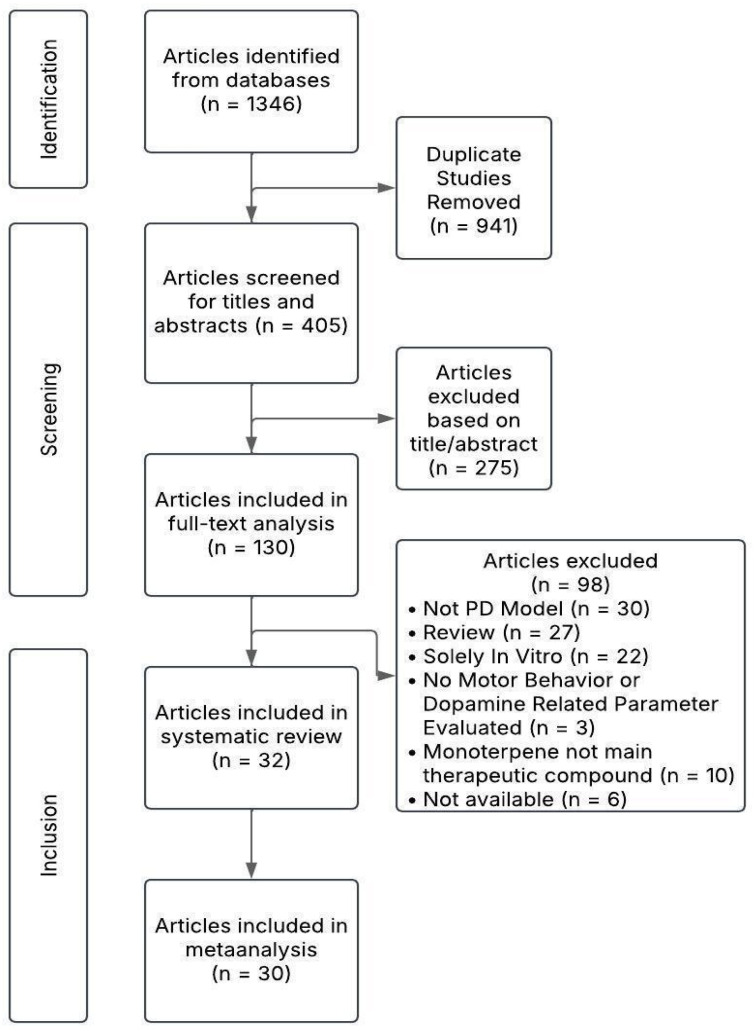
Flow diagram of literary search results.

**Figure 2 plants-14-00999-f002:**
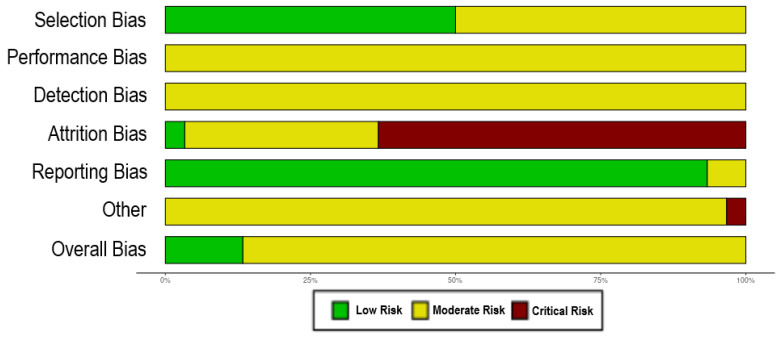
Risk of bias assessment. Summary of the risk of bias evaluation across the included studies in the meta-analysis. Six types of bias were addressed according to the SYRCLE’s Risk of Bias tool, and an overall bias was determined. Each domain is represented with a bar for the total number of publications analyzed. The green portion of the bar represents percentage of publications with low risk of bias, the yellow portion represents percentage of publications with moderate risk of bias, and the red portion represents percentage of publications with critical risk of bias.

**Figure 3 plants-14-00999-f003:**
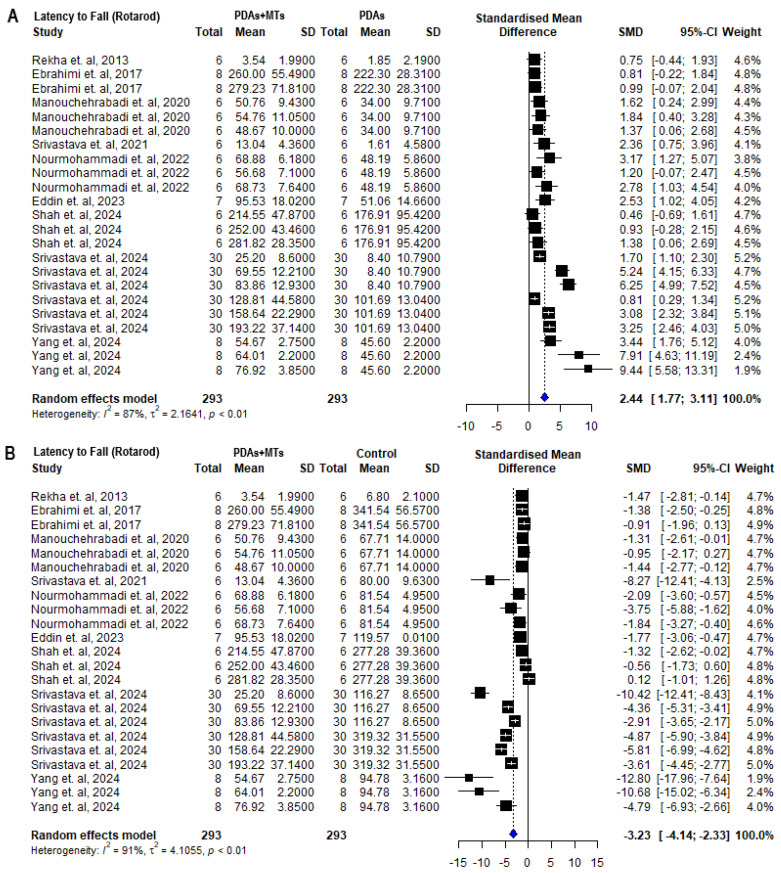
Forest plot comparing time to fall measurements in the Rotarod behavioral test of (**A**) parkinsonian animals (PDAs) versus PDAs treated with MT therapy (PDAs + MTs), and (**B**) PDAs + MTs versus control animals. Effect size is reported as standard mean deviation (SMD), and the variance is reported as the 95% confidence interval (CI). Individual study estimates are represented by squares, with their size reflecting study weight in the meta-analysis, and horizontal lines indicating 95% CIs. The diamond represents the overall pooled effect and the vertical line denotes the line of no effect (SMD = 0). A negative SMD represents less time to fall, whereas a positive SMD represents more time to fall. (**A**) shows a significant overall increase in the time PDAs remain on the Rotarod following treatment with MTs. (**B**) illustrates that the overall analysis reveals significant differences between PDAs and control animals, with the latter spending more time on the Rotarod [34,36,38,41,49,50,51,53,56].

**Figure 4 plants-14-00999-f004:**
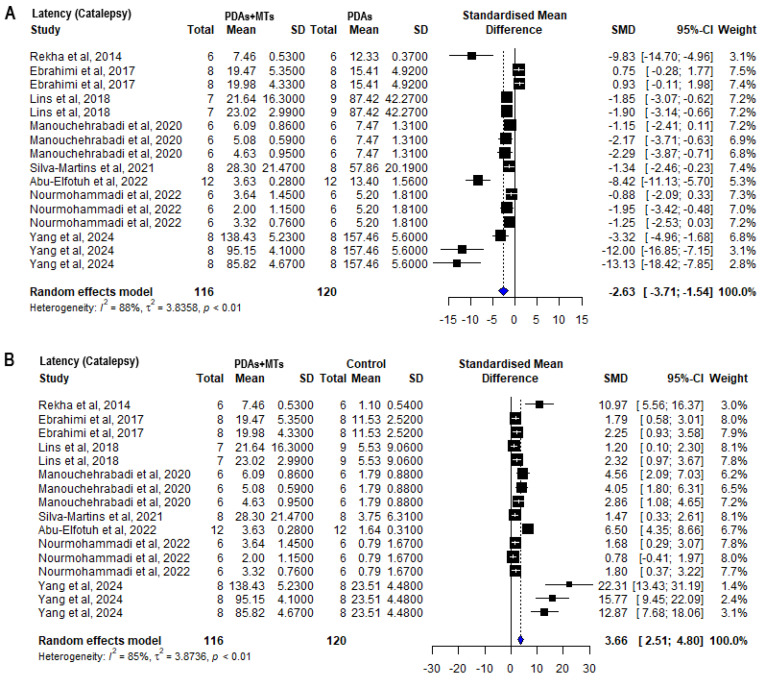
Forest plot comparing latency measurements in Catalepsy behavioral test of (**A**) parkinsonian animals (PDAs) versus PDAs treated with a MT therapy (PDAs + MTs), and (**B**) PDAs + MTs versus control animals. Effect size is reported as standard mean deviation (SMD), and the variance is reported as the 95% confidence interval (CI). Individual study estimates are represented by squares, with their size reflecting study weight in the meta-analysis, and horizontal lines indicating 95% CIs. The diamond represents the overall pooled effect and the vertical line denotes the line of no effect (SMD = 0). A negative SMD represents less time to move, whereas a positive SMD represents more time to move. (**A**) shows a significant overall decrease in the time PDAs took to move following treatment with MTs. (**B**) illustrates that the overall analysis reveals significant differences between PDAs and control animals, with the latter taking less time to move [33,34,40,47,51,53,54,56].

**Figure 5 plants-14-00999-f005:**
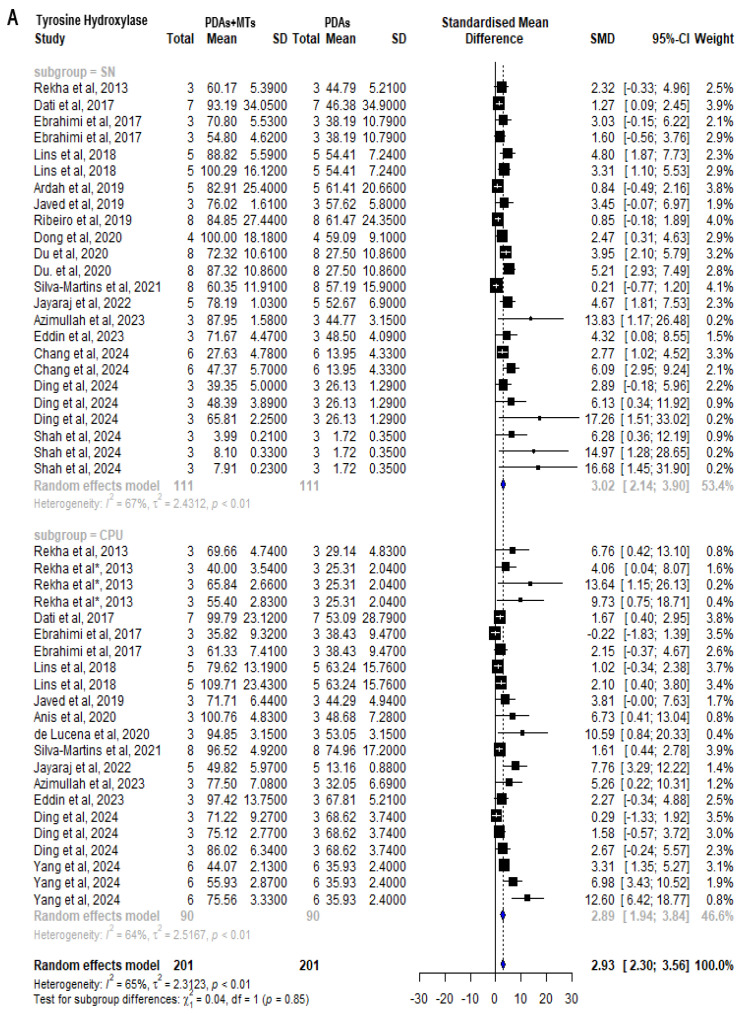
Forest plot comparing tyrosine hydroxylase (TH) levels in substantia nigra (SN) and striatum (CPU) of (**A**) parkinsonian animals (PDAs) versus PDAs treated with MT therapy (PDAs + MTs), and (**B**) PDAs + MTs versus control animals. Effect size is reported as standard mean deviation (SMD), and the variance is reported as the 95% confidence interval (CI). Individual study estimates are represented by squares, with their size reflecting study weight in the meta-analysis, and horizontal lines indicating 95% CIs. The diamond represents the overall pooled effect and the vertical line denotes the line of no effect (SMD = 0). A negative SMD represents lower TH levels, whereas a positive SMD represents higher TH levels. (**A**) shows a significant overall increase in TH content in SN and CPU of PDAs following treatment with MTs. (**B**) illustrates that the overall analysis reveals significant differences between PDAs and control animals, with the latter having higher levels of TH in SN and CPU [27,30,33,35,36,37,38,39,41,42,43,44,45,47,48,51,52,56,57,58].

**Figure 6 plants-14-00999-f006:**
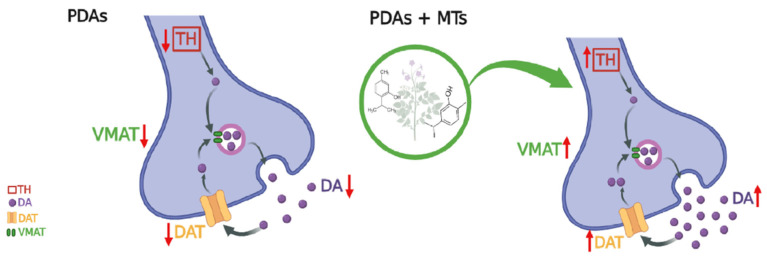
Representation of main changes in dopamine-related regulation observed in Parkinson’s disease animal models reporting the hypothesized effects of monoterpene administration. It should be noted that these potential mechanisms are derived from animal model studies and are largely speculative. The findings presented here cannot be directly translated to human Parkinson’s disease patients without further clinical research. PDAs (Parkinson’s disease animals); MTs (monoterpenes); TH (tyrosine hydroxylase); DA (dopamine); DAT (dopamine active transporter); VMAT (vesicular monoamine transporter).

**Figure 7 plants-14-00999-f007:**
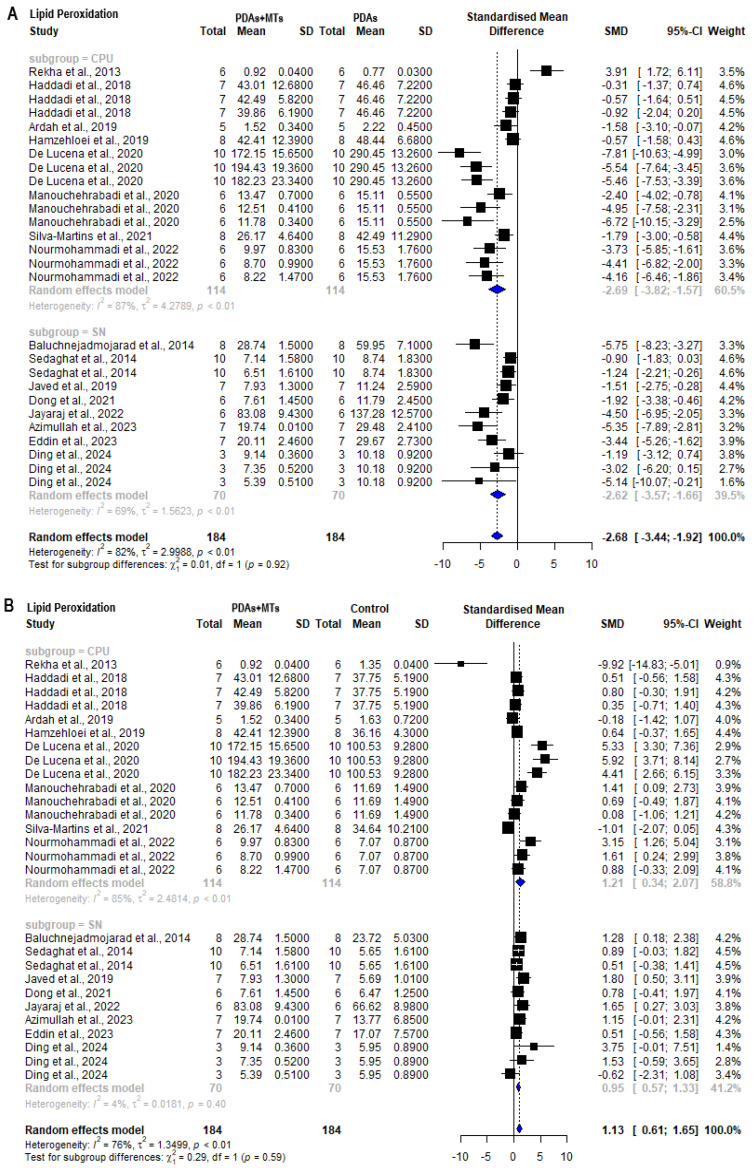
Forest plot comparing malondialdehyde (MDA) levels in substantia nigra (SN) and striatum (CPU) of (**A**) parkinsonian animals (PDAs) versus PDAs treated with a MT therapy (PDAs + MTs), and (**B**) PDAs + MTs versus control animals. Effect size is reported as standard mean deviation (SMD), and the variance is reported as the 95% confidence interval (CI). Individual study estimates are represented by squares, with their size reflecting study weight in the meta-analysis, and horizontal lines indicating 95% CIs. The diamond represents the overall pooled effect and the vertical line denotes the line of no effect (SMD = 0). A negative SMD represents lower MDA levels, whereas a positive SMD represents higher MDA levels. (**A**) shows a significant overall decrease in MDA levels in SN and CPU of PDAs following treatment with MTs. (**B**) illustrates that the overall analysis reveals significant differences between PDAs and control animals, with the latter having lower levels of MDA in SN and CPU [27,29,31,32,34,37,38,41,42,45,47,52,53,55,57,58].

**Figure 8 plants-14-00999-f008:**
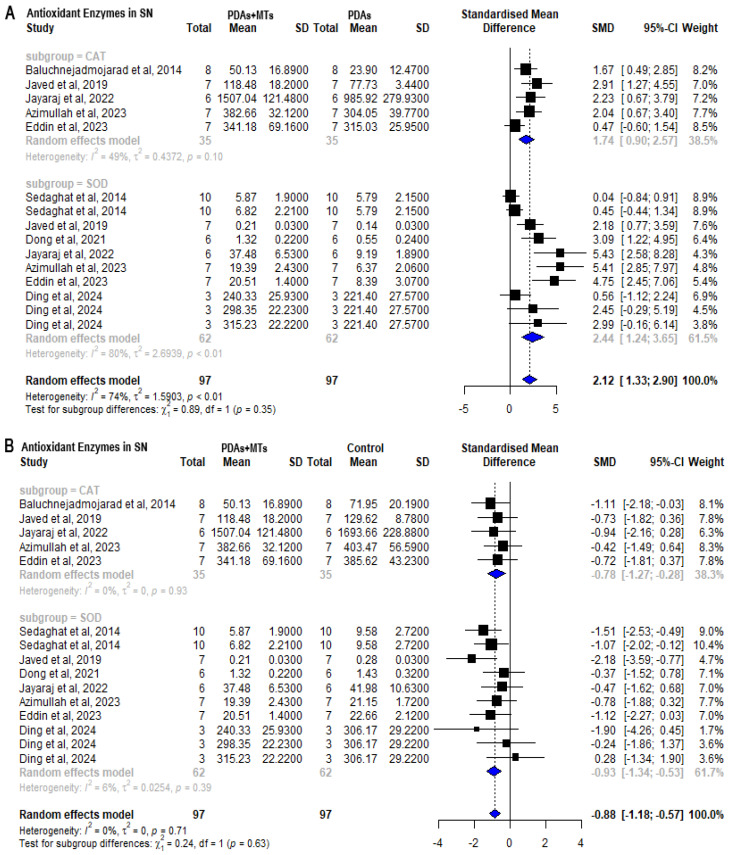
Forest plot comparing antioxidant enzymes’ (superoxide dismutase (SOD) and catalase (CAT)) activity/levels in substantia nigra (SN) of (**A**) parkinsonian animals (PDAs) versus PDAs treated with MT therapy (PDAs + MTs), and (**B**) PDAs + MTs versus control animals. Effect size is reported as standard mean deviation (SMD), and the variance is reported as the 95% confidence interval (CI). Individual study estimates are represented by squares, with their size reflecting study weight in the meta-analysis, and horizontal lines indicating 95% CIs. The diamond represents the overall pooled effect and the vertical line denotes the line of no effect (SMD = 0). A negative SMD represents lower levels or activity of antioxidant enzymes, whereas a positive SMD represents higher levels or activity of antioxidant enzymes. (**A**) shows a significant overall increase in antioxidant enzymes activity/levels in SN of PDAs following treatment with MTs. (**B**) illustrates that the overall analysis reveals significant differences between PDAs and control animals, with the latter having higher levels of antioxidant enzymes’ activity/levels in SN [27,29,37,41,45,52,55,58].

**Figure 9 plants-14-00999-f009:**
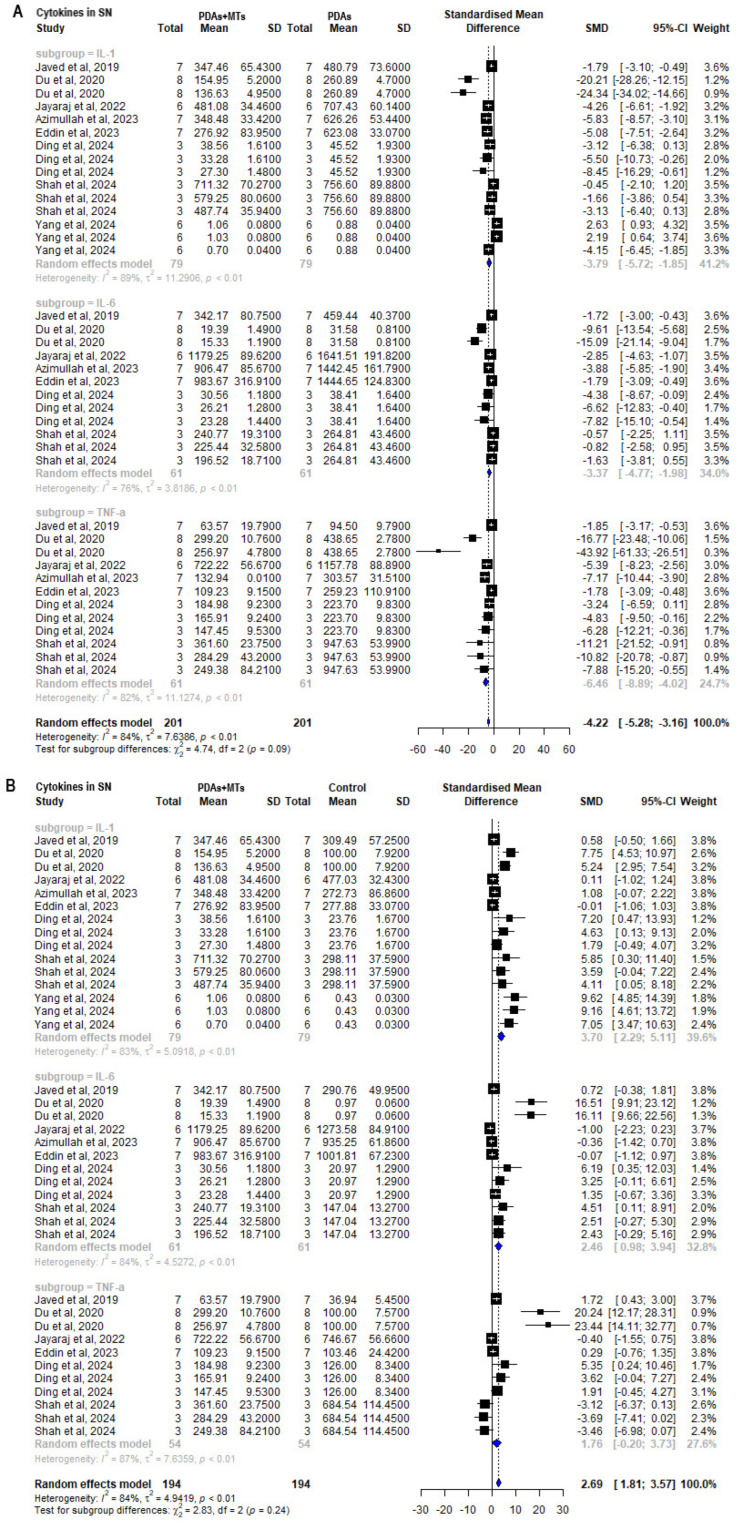
Forest plot comparing cytokines’ (interleukin-1 beta (IL-1β), interleukin-6 (IL-6), and tumor necrosis factor (TNF-α)) levels in substantia nigra (SN) of (**A**) parkinsonian animals (PDAs) versus PDAs treated with a MT therapy (PDAs + MTs), and (**B**) PDAs + MTs versus control animals. Effect size is reported as standard mean deviation (SMD), and the variance is reported as the 95% confidence interval (CI). Individual study estimates are represented by squares, with their size reflecting study weight in the meta-analysis, and horizontal lines indicating 95% CIs. The diamond represents the overall pooled effect and the vertical line denotes the line of no effect (SMD = 0). A negative SMD represents lower cytokine levels, whereas a positive SMD represents higher cytokine levels. (**A**) shows a significant overall decrease in cytokines’ levels in SN of PDAs following treatment with MTs. (**B**) illustrates that the overall analysis reveals significant differences between PDAs and control animals for IL-1β and IL-6, with the latter having higher levels of antioxidant enzymes’ activity/levels in SN, but no significant differences for TNF-α [27,36,37,41,44,45,51,52].

## Data Availability

The data supporting this study’s findings are presented within the article.

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
