# Peer review of "Plant-Derived Monoterpene Therapies in Parkinson’s Disease Models: Systematic Review and Meta-Analysis"

_plants, 2025, doi:10.3390/plants14070999_

Round 1

Reviewer 1 Report

Comments and Suggestions for Authors

This manuscript have been revised and eligible to publish after minor revision (see in the manuscript)

Comments on the Quality of English Language

The quality of English language is good enough

Author Response

Response to Reviewer 1

We sincerely appreciate your corrections and valuable suggestions.

Regarding your questions: Unfortunately, we were unable to include SciFinder in our analysis, as we do not have access to this resource.

As for the table format, the original file we submitted does not contain extra spaces; it is possible that modifications occurred during the journal's editing process.

With these comments addressed, all the other recommended revisions have been implemented in the text.

Reviewer 2 Report

Comments and Suggestions for Authors

The introduction provides a solid foundation on PD pathophysiology and the potential of MTs as neuroprotective agents. However, it should further emphasize the rationale for conducting a meta-analysis and justify why monoterpenes, rather than other plant secondary metabolites, were selected. Additionally, the background on PRISMA methodology could be clarified.

The systematic review and meta-analysis methodology is well-structured, and the selection of studies follows a rigorous inclusion/exclusion process. The risk of bias assessment enhances credibility.

The methodology needs greater clarity on statistical models: (1) How heterogeneity (I²) was managed across studies should be explicitly explained. (2) Details on how the meta-analysis weighted different studies (e.g., fixed vs. random effects models) should be clearly stated. (3) Search strategy and study selection criteria could be more concisely summarized.

The results are well-organized, but several figures (meta-analysis forest plots, risk of bias assessment) require clearer legends and better descriptions. Some redundant text could be streamlined to enhance readability. Additionally, the limitations section should discuss publication bias more explicitly.

The conclusions are generally well-supported, but they should better differentiate between statistically significant findings and their biological relevance. It would be helpful to discuss potential mechanisms of action of MTs beyond antioxidant and anti-inflammatory effects, such as mitochondrial protection or neurotransmitter modulation.

Comments on the Quality of English Language

While scientifically sound, the text contains grammatical inconsistencies and overly complex sentence structures that make it difficult to follow in some sections. Improvements in clarity, conciseness, and technical precision are recommended. A professional English editing service could be beneficial.

Author Response

Response to Reviewer 2

Thank you very much for taking the time to review this manuscript. Please find the detailed responses below and the corresponding revisions/corrections in red in the re-submitted files.

Comments 1: The introduction provides a solid foundation on PD pathophysiology and the potential of MTs as neuroprotective agents. However, it should further emphasize the rationale for conducting a meta-analysis and justify why monoterpenes, rather than other plant secondary metabolites, were selected. Additionally, the background on PRISMA methodology could be clarified.

Response 1: Thank you for your comment. We added some sentences and modified the structure of the fourth paragraph of the introduction according to the suggestions. The background on PRISMA methodology will be included in the Supplementary material (Annex 1).

Comments 2: The methodology needs greater clarity on statistical models: (1) How heterogeneity (I²) was managed across studies should be explicitly explained. (2) Details on how the meta-analysis weighted different studies (e.g., fixed vs. random effects models) should be clearly stated. (3) Search strategy and study selection criteria could be more concisely summarized.

Response 2: Thank you for your comment. We added the criteria on I² values to explain heterogeneity, and the model of meta-analysis (random-effects model) in the methodology section. Search strategy and study selection criteria were removed from the methodology section and will be included in the Supplementary material (Annex 1), along with the PRISMA methodology background.

Comments 3: The results are well-organized, but several figures (meta-analysis forest plots, risk of bias assessment) require clearer legends and better descriptions. Some redundant text could be streamlined to enhance readability. Additionally, the limitations section should discuss publication bias more explicitly.

Response 3: Thank you for your valuable comment. We made sure that the figure legends already include the requested clarifications, providing detailed explanations. We have also expanded the limitations section (2.8. Limitations section) to explicitly discuss publication bias.

Comments 4: The conclusions are generally well-supported, but they should better differentiate between statistically significant findings and their biological relevance. It would be helpful to discuss potential mechanisms of action of MTs beyond antioxidant and anti-inflammatory effects, such as mitochondrial protection or neurotransmitter modulation.

Response 4: Thank you for your appropriate comment to improve the manuscript. We have taken into account the comments and modified the manuscript accordingly. We have included the following text to the manuscript for mitochondrial protection:

“Although not included in the meta-analysis due to the limited number of measurements, numerous studies have reported the effects of MTs administration on ……… mitochondrial integrity, which is closely linked to apoptosis, in brain cells. Most neurotoxic compounds used to induce Parkinsonian symptoms in animal models target mitochondria, causing instability and increasing oxidant species production, which drive the neurodegenerative process. Mitochondrial markers assessed in multiple studies indicate that MT administration mitigates mitochondrial dysfunction in PD models (Rekha et al, 2014; Eddin et al, 2023; Ebrahimi et al, 2017; Anis et al, 2020).”

Rekha, K. R., & Selvakumar, G. P. (2014). Gene expression regulation of Bcl2, Bax and cytochrome-C by geraniol on chronic MPTP/probenecid induced C57BL/6 mice model of Parkinson’s disease. Chemico-biological interactions, 217, 57-66.

Eddin, L. B., Azimullah, S., Jha, N. K., Nagoor Meeran, M. F., Beiram, R., & Ojha, S. (2023). Limonene, a monoterpene, mitigates rotenone-induced dopaminergic neurodegeneration by modulating neuroinflammation, hippo signaling and apoptosis in rats. International journal of molecular sciences, 24(6), 5222.

Ebrahimi, S. S., Oryan, S., Izadpanah, E., & Hassanzadeh, K. (2017). Thymoquinone exerts neuroprotective effect in animal model of Parkinson’s disease. Toxicology letters, 276, 108-114.

Anis, E., Zafeer, M. F., Firdaus, F., Islam, S. N., Khan, A. A., & Hossain, M. M. (2020). Perillyl alcohol mitigates behavioural changes and limits cell death and mitochondrial changes in unilateral 6-OHDA lesion model of Parkinson’s disease through alleviation of oxidative stress. Neurotoxicity research, 38, 461-477.

We have included the following text to the manuscript for neurotransmitter modulation:

“These results demonstrated that MTs can also have anxiolytic, antidepressant, memory-enhancing and sensorial effects, likely through the modulation of glutamate and gamma-aminobutyric acid neurotransmitter systems, as reported by Agatonovic-Kustrin et al, 2020”.

Agatonovic-Kustrin, S., Kustrin, E., Gegechkori, V., & Morton, D. W. (2020). Anxiolytic terpenoids and aromatherapy for anxiety and depression. Reviews on New Drug Targets in Age-Related Disorders, 283-296.

Comments 5: Comments on the Quality of English Language

While scientifically sound, the text contains grammatical inconsistencies and overly complex sentence structures that make it difficult to follow in some sections. Improvements in clarity, conciseness, and technical precision are recommended. A professional English editing service could be beneficial.

Response 5: The English language of the manuscript has been revised by M.J. Martinez, Sworn Translator of English (Professional License No. 532)

Reviewer 3 Report

Comments and Suggestions for Authors

The manuscript entitled "Plant-Derived Monoterpene Therapies in Parkinson's Disease Models: Systematic Review and Meta-Analysis" is a systematic review and a meta-analysis on the use of monoterpenes in the treatment of Parkinson in animal models. The focus of this manuscript is certainly of interest, although there are some points that need to be addressed. The following suggestions could potentially add value to the work:

  1. Why did the authors specifically include murine models? Are there other types of Parkinson's disease animal models?
  2. It would be useful to introduce the differences between animal models and Parkinson's disease in humans, reporting the limitations of studies on animals.
  3. 114 “All models of Parkinsonism were induced chemically.” Are there other in vivo models of Parkinson's disease?
  4. 175 Regarding the sentence: "Most reports demonstrated improvements in evaluated motor parameters following MTs treatment (Table 1)." Were the improvements statistically significant in each study? What range of effectiveness was detected?
  5. Did all monoterpenes protect in the same manner? Why did the authors consider them all together?
  6. Figure 6 illustrates graphically the changes in dopamine-related parameters in Parkinson's disease animal models and the effects of monoterpene administration. While the figure is useful, the caption has to be improved to ensure clarity and prevent potential misinterpretation. An example of how change it is the following:

Figure 6’s caption: Representation of main changes in dopamine-related regulation observed in Parkinson's disease animal models reporting the hypothesized effects of monoterpene administration. It should be noted that these potential mechanisms are derived from animal model studies and are largely speculative. The findings presented here cannot be directly translated to human Parkinson's disease patients without further clinical research. PDAs (Parkinson's disease Animals); MTs (monoterpenes); TH (tyrosine hydroxylase); DA (dopamine); DAT (dopamine active transporter); VMAT (vesicular monoamine transporter).

  1. 365-367 Regarding the sentence: “In the Rotarod test, most studies reported that PDAs spent significantly more time in the accelerating spinning rod than PDAs+ MTs (Figure 3), typically with doses between 25 and 100 mg/kg.” This statement appears to be either incorrect or unclear. Typically, in Rotarod tests, treated animals (in this case, PDAs+ MTs) are expected to perform better, spending more time on the accelerating rod compared to untreated animals (PDAs). The current wording suggests the opposite, which seems counterintuitive given the context of the study. Could you please verify if this sentence accurately reflects the results of the study? If the statement is correct as written, it would be helpful to provide additional explanation or context to clarify why the untreated animals performed better in this test, as this would be an unexpected outcome.
  2. 367-368 About the sentence: “However, some MTs have enhanced motor performance at much lower concentrations, as low as 0.1 mg/kg [47].” Can the authors specify which type of MT it is?
  3. While this meta-analysis presents some valuable insights, the methodology raises significant concerns. The analysis treats various monoterpenes as a homogeneous entity, which is a flawed assumption given the potential variability in their chemical structures and pharmacological activities. This approach may obscure important differences in efficacy and mechanism of action among individual compounds. To address this limitation, it is recommended conducting sub-analyses that focus on the most active monoterpenes individually. This would provide a more accurate and scientifically rigorous assessment of their respective therapeutic potentials in Parkinson's disease models. Such an approach would not only enhance the validity of the study, but also offer more targeted directions for future research and potential clinical applications.
  4. 462-463 The dates are written in different formats. It is recommended to standardize the date format for consistency and clarity.
  5. Please check all the acronyms and ensure that all are explained at their first mention.

Author Response

Response to Reviewer 3

Thank you very much for taking the time to review this manuscript. Please find the detailed responses below and the corresponding revisions/corrections in red in the re-submitted files.

Comments 1: The manuscript entitled "Plant-Derived Monoterpene Therapies in Parkinson's Disease Models: Systematic Review and Meta-Analysis" is a systematic review and a meta-analysis on the use of monoterpenes in the treatment of Parkinson in animal models. The focus of this manuscript is certainly of interest, although there are some points that need to be addressed. The following suggestions could potentially add value to the work: Why did the authors specifically include murine models? Are there other types of Parkinson's disease animal models?It would be useful to introduce the differences between animal models and Parkinson's disease in humans, reporting the limitations of studies on animals.

Response 1: Thank you for your comment. Indeed, other animal models of Parkinson’s disease exist, such as Drosophila melanogaster (Victor Atoki A. et al., 2025) and Caenorhabditis elegans (Parrales et al., 2025), among others. However, murine models are the most widely used in scientific literature, as they exhibit behavioral and biochemical alterations during both the prodromal phase and disease progression that more closely resemble the neurodegenerative process in humans.

Victor Atoki A, Aja PM, Shinkafi TS, Ondari EN, Adeniyi AI, Fasogbon IV, Dangana RS, Shehu UU, Akin-Adewumi A.Fly (Austin). 2025 Dec;19(1):2420453.   Exploring the versatility of Drosophila melanogaster as a model organism in biomedical research: a comprehensive review. doi: 10.1080/19336934.2024.2420453

Valeria Parrales, Guillaume Arcile , Louise Laserre, Sébastien Normant , Géraldine Le Goff , Christian Da Costa Noble, Jamal Ouazzani , Noelle Callizot , Stéphane Haïk , Chérif Rabhi , Nicolas Bizat . Neuroprotective Effect of Withaferin Derivatives toward MPP+ and 6-OHDA Toxicity to Dopaminergic Neurons ACS Chem Neurosci. 2025 Mar 5;16(5):802-817. doi: 10.1021/acschemneuro.4c00655. Epub 2025 Feb 13.

In response to the reviewer's valuable comment, we have included the following text to the manuscript:

“Murine models of PD are widely used to study the neurodegenerative nigrostriatal process (Ahmad et al., 2005), as they exhibit behavioral, biochemical, and molecular neuroinflammatory alterations like those observed in humans. These models also demonstrate dopaminergic neuron loss driven by the formation of various oxidants and free radicals, neuroinflammation, lipid peroxidation, and the depletion of reduced glutathione. Despite its limitations in brain structures, murine models remain a valuable tool for investigating the pathogenesis and progression of PD (Fasano, 2012)”.

 Ahmad, A. S., Ansari, M. A., Ahmad, M., Saleem, S., Yousuf, S., Hoda, M. N., & Islam, F. (2005). Neuroprotection by crocetin in a hemi-parkinsonian rat model. Pharmacology Biochemistry and Behavior, 81(4), 805-813.

Fasano, M. (2012). Biochemistry of Parkinson's disease--insights from cellular models, animal models and human tissue specimens obtained by autopsy. The FEBS Journal, 279(7), 1145-1145.

Comments 2: 114 “All models of Parkinsonism were induced chemically.” Are there other in vivo models of Parkinson's disease?

Response 2:  Thank you for pointing this out. Most models are induced chemically as reported in the results section. However, there are genetic-based models that include transgenic animals that contain mutated genes associated with PD and animals that overexpress proteins such as α-synuclein through viral vectors. In the systematic review we took these models into consideration when we constructed the search protocol, as can be seen in the PROSPERO database (https://www.crd.york.ac.uk/prospero/display_record.php?ID=CRD42024592555). However, we did not find any publication that analyzed monoterpene administration on a genetic model of PD; therefore, they were not discussed.

Considering the editor’s feedback, the sentence has been modified in the text for greater clarity.

Comments 3: 175 Regarding the sentence: "Most reports demonstrated improvements in evaluated motor parameters following MTs treatment (Table 1)." Were the improvements statistically significant in each study? What range of effectiveness was detected?

Response 3:  In Table 1, all changes that are represented with arrows report a statistically significant result in PDA+MTs compared with PDAs. This was modified in the legend of the table for more clarity. On the other hand, most included studies did not specify the range of effectiveness.

Comments 4: Did all monoterpenes protect in the same manner? Why did the authors consider them all together?

Response 4:  Thank you for your insightful comment.

Although MTs constitute a diverse group of compounds with distinct chemical compositions, they are widely recognized for their antioxidant and anti-inflammatory properties, despite differences in their mechanisms of action. Moreover, their biological activity depends on factors such as dosage, duration, route of administration, and the specific biological system in which they are tested. Nevertheless, MTs share certain physicochemical characteristics that contribute to their similar biological properties. Therefore, all MTs included in this review were analyzed collectively to provide a comprehensive evaluation of their overall effects.

In response to the reviewer's valuable comment, we have included the following text to the manuscript:

“Although MTs comprise a diverse group of compounds with distinct chemical compositions, they share certain physicochemical characteristics that contribute to their similar biological properties. Therefore, all MTs included in this review were analyzed collectively to provide a comprehensive evaluation of their overall effects”.

Comments 5: Figure 6 illustrates graphically the changes in dopamine-related parameters in Parkinson's disease animal models and the effects of monoterpene administration. While the figure is useful, the caption has to be improved to ensure clarity and prevent potential misinterpretation. An example of how change it is the following:

Figure 6’s caption: Representation of main changes in dopamine-related regulation observed in Parkinson's disease animal models reporting the hypothesized effects of monoterpene administration. It should be noted that these potential mechanisms are derived from animal model studies and are largely speculative. The findings presented here cannot be directly translated to human Parkinson's disease patients without further clinical research. PDAs (Parkinson's disease Animals); MTs (monoterpenes); TH (tyrosine hydroxylase); DA (dopamine); DAT (dopamine active transporter); VMAT (vesicular monoamine transporter).

Response 5:  Thank you very much for your suggestion. We have improved the caption to clarify and prevent potential misinterpretation.

Comments 6: 365-367 Regarding the sentence: “In the Rotarod test, most studies reported that PDAs spent significantly more time in the accelerating spinning rod than PDAs+ MTs (Figure 3), typically with doses between 25 and 100 mg/kg.” This statement appears to be either incorrect or unclear. Typically, in Rotarod tests, treated animals (in this case, PDAs+ MTs) are expected to perform better, spending more time on the accelerating rod compared to untreated animals (PDAs). The current wording suggests the opposite, which seems counterintuitive given the context of the study. Could you please verify if this sentence accurately reflects the results of the study? If the statement is correct as written, it would be helpful to provide additional explanation or context to clarify why the untreated animals performed better in this test, as this would be an unexpected outcome.

Response 6:  Thank you for noticing this error. We appreciate your careful review, and we have now corrected it in the manuscript.

Comments 7: 367-368 About the sentence: “However, some MTs have enhanced motor performance at much lower concentrations, as low as 0.1 mg/kg [47].” Can the authors specify which type of MT it is?

Response 7:   The monoterpene referred to is safranal, a cyclic oxygenated monoterpene with an aldehyde functional group and is now specified in the text. Currently correspond to the reference #51

Comments 8: While this meta-analysis presents some valuable insights, the methodology raises significant concerns. The analysis treats various monoterpenes as a homogeneous entity, which is a flawed assumption given the potential variability in their chemical structures and pharmacological activities. This approach may obscure important differences in efficacy and mechanism of action among individual compounds. To address this limitation, it is recommended conducting sub-analyses that focus on the most active monoterpenes individually. This would provide a more accurate and scientifically rigorous assessment of their respective therapeutic potentials in Parkinson's disease models. Such an approach would not only enhance the validity of the study, but also offer more targeted directions for future research and potential clinical applications.

Response 8: Thank you for this pertinent comment. Unfortunately, the available literature does not provide enough studies to perform a comprehensive subgroup analysis for each individual monoterpene. However, we conducted a preliminary subgroup analysis conducted for the reviewer using the most robust behavioral test, the Rotarod, based on the limited available data. Our results showed no statistically significant differences among most monoterpenes (Figure in PDF file). Only safranal exhibited significant differences compared to geraniol, carvacrol, and thymoquinone, though the sample size remains too small for a more representative analysis. Due to insufficient sample size, these findings were not included in the final manuscript.

We hope that the results of our work will inspire further research to better identify and characterize specific bioactive compounds with neuroprotective potential. 

Comments 9: 462-463 The dates are written in different formats. It is recommended to standardize the date format for consistency and clarity.

Response 9: Thank you for your observation. We have now standardized the date format in the manuscript.

Comments 10: Please check all the acronyms and ensure that all are explained at their first mention.

Response 10: Thank you for your suggestion. We have reviewed all acronyms and ensured that they are explained at their first mention in the manuscript.

Round 2

Reviewer 2 Report

Comments and Suggestions for Authors

I consider that the authors have covered all the comments and suggestions.

Reviewer 3 Report

Comments and Suggestions for Authors

Dear Authors,

I appreciated your revision where most critical points have been taken into account and resolved. I think that your review can be useful to several researchers, and for this reason, I suggest the publication of your manuscript.